# Anti-Oversmoothing in Deep Vision Transformers via the Fourier Domain Analysis: From Theory to Practice

**Peihao Wang, Wenqing Zheng, Tianlong Chen & Zhangyang Wang**
Department of Electrical and Computer Engineering, The University of Texas at Austin
`{peihaowang,w.zheng,tianlong.chen,atlaswang}@utexas.edu`

## Abstract

Vision Transformer (ViT) has recently demonstrated promise in computer vision problems. However, unlike Convolutional Neural Networks (CNN), it is known that the performance of ViT saturates quickly with depth increasing, due to the observed attention collapse or patch uniformity. Despite a couple of empirical solutions, a rigorous framework studying on this scalability issue remains elusive. In this paper, we first establish a **rigorous theory framework** to analyze ViT features from the Fourier spectrum domain. We show that the self-attention mechanism inherently amounts to a low-pass filter, which indicates when ViT scales up its depth, excessive low-pass filtering will cause feature maps to only preserve their Direct-Current (DC) component. We then propose two straightforward yet effective techniques to mitigate the undesirable low-pass limitation. The first technique, termed **AttnScale**, decomposes a self-attention block into low-pass and high-pass components, then rescales and combines these two filters to produce an all-pass self-attention matrix. The second technique, termed **FeatScale**, re-weights feature maps on separate frequency bands to amplify the high-frequency signals. Both techniques are efficient and hyperparameter-free, while effectively overcoming relevant ViT training artifacts such as attention collapse and patch uniformity. By seamlessly plugging in our techniques to multiple ViT variants, we demonstrate that they consistently help ViTs benefit from deeper architectures, bringing **up to 1.1% performance gains "for free"** (e.g., with little parameter overhead). We publicly release our codes and pre-trained models at `https://github.com/VITA-Group/ViT-Anti-Oversmoothing`.

## 1 Introduction

Transformers have achieved phenomenal success in Natural Language Processing (NLP) (Vaswani et al., 2017; Devlin et al., 2018; Dai et al., 2019; Brown et al., 2020), and recently in a wide range of computer vision applications too (Dosovitskiy et al., 2020; Liu et al., 2021; Arnab et al., 2021; Carion et al., 2020; Jiang et al., 2021a). One representative advance, the Vision Transformer (ViT) (Dosovitskiy et al., 2020), stacks Multi-head Self-Attention (MSA) blocks, by treating each local image patch as semantic tokens and modeling their interactions globally. Unlike Convolutional Neural Networks (CNNs) that hierarchically enlarge the receptive from local to global, even a shallow ViT is able to effectively capture the global contexts, leading to their very competitive performance on image classification and other tasks (Liu et al., 2021; Jiang et al., 2021a).

Going deep has always been a trend in deep learning (LeCun et al., 2015; Krizhevsky et al., 2012), and ViT was expected to make no exception. One might reasonably conjecture that a deeper ViT with more MSA blocks significantly outperform its shallower baseline. Unfortunately, building deeper ViTs face practical challenges. Empirically, Zhou et al. (2021a) shows a vanilla ViT of 32 layers under-performs the 24-layer one. Gong et al. (2021) demonstrates a downgraded patch diversity in deeper layers, and Dong et al. (2021) mathematically reveals the rank collapse phenomenon when Transformer goes deeper. Despite efforts towards deep ViT through patch diversification (Gong et al., 2021; Zhou et al., 2021b), rank collapse alleviation (Zhou et al., 2021a; Zhang et al., 2021), and training stabilization (Touvron et al., 2021b; Zhang et al., 2019), most of them are restricted to

empirical studies. Rethinking the problem with deep ViT from a more principled angle pends further efforts.

In this paper, we present the first rigorous analysis of stacking self-attention mechanism in the Fourier space. We mathematically show that cascading self-attention blocks is equivalent to repeatedly applying a low-pass filter, regardless of the input key or query tokens (Section 2.2). As a consequence, going deeper with vanilla ViT blocks only preserves Direct Component (DC) of the signal at the output layer. This theoretical finding explains the observations of patch uniformity and rank collapse, and is also inherently related to the over-smoothing phenomenon in Graph Convolutional Networks (GCNs) (Kipf & Welling, 2017; NT & Maehara, 2021; Oono & Suzuki, 2019; Cai & Wang, 2020). Moreover, we also reveal the role of other transformer modules (e.g., MLP and residual connection) in preventing this undesirable low-pass filtering (Section 2.3).

Built on the aforementioned analysis framework in the Fourier domain, we propose two novel techniques, to mitigate the low-pass filtering effect of self-attention and effectively scale up the depth of ViTs. The first technique, termed *Attention Scaling* (**AttnScale**), directly manipulates on the calculated attention map to enforce an all-pass filter (Section 3.1). It decomposes the self-attention matrix into a low-pass filter plus a high-pass filter, then adopts a learnable weight to adaptively amplify the effect of high-pass filter. The second technique, termed *Feature Scaling* (**FeatScale**), hinges on feature maps to re-weight different frequency bands separately (Section 3.2). It employs trainable coefficients to re-mix the DC and high-frequency components, hence selectively enhancing the high-frequency portion of the MSA output. Both AttnScale and FeatScale are extremely memory and computationally friendly. Neither runs Fourier transformation explicitly, bringing little extra complexity to the original ViTs.

Our contributions can be summarized as follows:

- We establish the first rigorous theoretical analysis of ViT from the spectral domain. We characterize the low-pass filtering effect of cascading MSAs, which connects to the recent empirical findings of ViT patch diversity loss or rank collapse.

- We present two theoretically grounded Fourier-domain scaling techniques, named AttnScale and FeatScale. They operating on re-adjusting the low- and high-frequency components of the attention maps and feature maps, respectively. Both are efficient, hyperparameter-free, easy-to-use, and able to generalize across different ViT variants.

- We conduct extensive experiments by integrating AttnScale and FeatScale with different ViT backbones. Both of our approaches substantially boost DeiT, CaiT, and Swin-Transformer with up to $1.1\%$, $0.6\%$ and $0.5\%$ performance gains, without whistles and bells.

## 2 WHY VIT CANNOT GO DEEPER?

### 2.1 NOTATION AND PRELIMINARIES

We begin by introducing our notations. Let $\boldsymbol{X} \in \mathbb{R}^{n \times d}$ denote the feature matrix, where $n$ is the number of samples, and $d$ is the feature dimension. Let $\boldsymbol{x}_i \in \mathbb{R}^d, \forall i = 1, \cdots, n$, the $i$-th row of $\boldsymbol{X}$, denote the feature vector of the $i$-th sample, and $\boldsymbol{z}_j \in \mathbb{R}^n, \forall j = 1, \cdots, d$, the $j$-th column of $\boldsymbol{X}$, represent signals of the $j$-th channel. In the context of ViT, $\boldsymbol{X}$ denotes a set (sequence) of image patches, $\boldsymbol{x}_i$ denotes the flatten version of the $i$-th patch embedding ($d$ = patch width $\times$ patch height), and $\boldsymbol{z}_j$ denotes a whole image signal of the $j$-th channel.

**Transformer Architecture** Vision Transformer (ViT) consists of three main components: a patch embedding and position encoding part, a stack of transformer encoder block with Multi-Head Self-Attention (MSA) and Feed-Forward Network (FFN), and a score readout function for image classification. We depict a transformer block in Fig. 3a. The key ingredient here is the Self-Attention (SA) module, which takes in the token representation of the last layer, and encodes each image token by aggregating information from other patches with respect to the computed attention value, formulated as below (Vaswani et al., 2017):

$$\text{SA}(\boldsymbol{X}) = \text{softmax}\left(\frac{\boldsymbol{X}\boldsymbol{W}_Q(\boldsymbol{X}\boldsymbol{W}_K)^T}{\sqrt{d}}\right)\boldsymbol{X}\boldsymbol{W}_V, \tag{1}$$

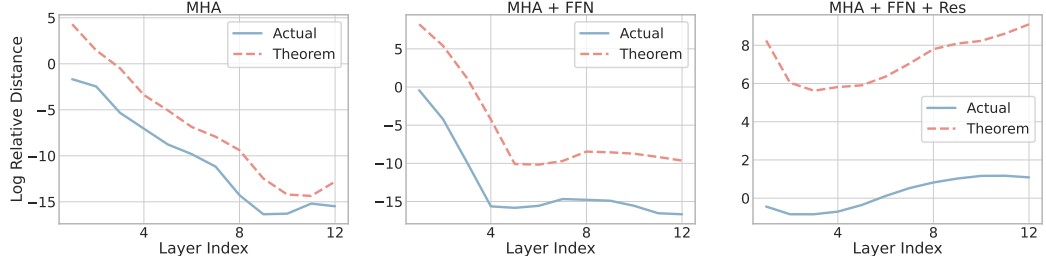

Figure 1: Visualize the intensity of high-frequency component and their theoretical upper bounds under different transformer blocks. The blue line is defined by $\log(\|\mathcal{HC}[\boldsymbol{X}_l]\|_F/\|\mathcal{HC}[\boldsymbol{X}_0]\|_F)$, and the red line is estimated using the results in Section 2.2 & 2.3. See details in Appendix F.1.

where $\boldsymbol{W}_K \in \mathbb{R}^{d \times d_k}, \boldsymbol{W}_Q \in \mathbb{R}^{d \times d_q}, \boldsymbol{W}_V \in \mathbb{R}^{d \times d}$ are the key, query, and value weight matrices, respectively, $\sqrt{d}$ here denotes a scaling factor, and $\mathrm{softmax}(\cdot)$ operates on $\boldsymbol{X}$ row-wisely. Multi-Head Self-Attention (MSA) involves a group of SA heads and combines their outputs through a linear projection (Vaswani et al., 2017):

$$\mathrm{MSA}(\boldsymbol{X}) = [\mathrm{SA}_1(\boldsymbol{X}) \quad \cdots \quad \mathrm{SA}_H(\boldsymbol{X})] \, \boldsymbol{W}_O, \tag{2}$$

where the subscripts denote the SA head number, $H$ is the total number of SA heads, and $\boldsymbol{W}_O \in \mathbb{R}^{Hd \times d}$ projects multi-head outputs to the hidden dimension. Besides MSA module, each transformer block is equipped with a normalization layer, feed-forward network, and skip connections to cooperate with MSA. Formally, a transformer block can be written as follows:

$$\boldsymbol{X}' = \mathrm{MSA}(\mathrm{LayerNorm}(\boldsymbol{X})) + \boldsymbol{X}, \tag{3}$$

$$\boldsymbol{Y} = \mathrm{FFN}(\mathrm{LayerNorm}(\boldsymbol{X}')) + \boldsymbol{X}'. \tag{4}$$

**Fourier Analysis** The main analytic tool in this paper is Fourier transform. Denote $\mathcal{F} : \mathbb{R}^n \to \mathbb{C}^n$ be the Discrete Fourier Transform (DFT) with the Inverse Discrete Fourier Transform (IFT) $\mathcal{F}^{-1} : \mathbb{C}^n \to \mathbb{R}^n$. Applying $\mathcal{F}$ to a flatten image signal $\boldsymbol{x}$ is equivalent to left multiplying a DFT matrix, whose rows are the Fourier basis $\boldsymbol{f}_k = \left[e^{2\pi \mathrm{j}(k-1)\cdot 0} \quad \cdots \quad e^{2\pi \mathrm{j}(k-1)\cdot(n-1)}\right]^T / \sqrt{n} \in \mathbb{R}^n$, where $k$ denotes the $k$-th row of DFT matrix, and $\mathrm{j}$ is the imaginary unit. Let $\tilde{\boldsymbol{z}} = \mathcal{F}\boldsymbol{z}$ be the spectrum of $\boldsymbol{z}$, and $\tilde{\boldsymbol{z}}_{dc} \in \mathbb{C}$, $\tilde{\boldsymbol{z}}_{hc} \in \mathbb{C}^{n-1}$ take the first element and the rest elements of $\tilde{\boldsymbol{z}}$, respectively. Define $\mathcal{DC}[\boldsymbol{z}] = \tilde{\boldsymbol{z}}_{dc}\boldsymbol{f}_1 \in \mathbb{C}^n$ as the Direct-Current (DC) component of signal $\boldsymbol{z}$, and $\mathcal{HC}[\boldsymbol{z}] = [\boldsymbol{f}_2 \quad \cdots \quad \boldsymbol{f}_n]\tilde{\boldsymbol{z}}_{hc} \in \mathbb{C}^n$ the complementary high-frequency component.

In signal processing, a low-pass filter is a system that suppresses the high-frequency component of signals and retain the low-frequency component. In this paper, we refer to low-pass filter as a particular type of filters that only preserve the DC component, while diminishing the remaining high-frequency component. To be more precise, we define low-pass filters in Definition 1.

**Definition 1.** *Given an endomorphism $f : \mathbb{R}^n \to \mathbb{R}^n$ with $f^t$ denoting applying $f$ for $t$ times, $f$ is a low-pass filter if and only if for all $\boldsymbol{z} \in \mathbb{R}^n$:*

$$\lim_{t \to \infty} \frac{\|\mathcal{HC}[f^t(\boldsymbol{z})]\|_2}{\|\mathcal{DC}[f^t(\boldsymbol{z})]\|_2} = 0. \tag{5}$$

Definition 1 reveals the nature of low-pass filters: they will produce a dominant response on DC component, while imposing an inhibition effect on the high-frequency band. We refer interested readers to Appendix A for more useful backgrounds.

## 2.2 SELF-ATTENTION IS A LOW-PASS FILTER

In this subsection, we will give theoretical justification on self-attention in terms of its spectral-domain effect. Our main result is that self-attention is constantly a low-pass filter, which continuously erases high-frequency information, thus causing ViT to lose features expressiveness at deep layers.

Formally, we have the following theorem that shows attention matrix produced by a softmax function (e.g, Eqn. 1) is a low-pass filter independent of the input token features or key/query matrices.

**Theorem 1.** *(SA matrix is a low-pass filter) Let $\boldsymbol{A} = \mathrm{softmax}(\boldsymbol{P})$, where $\boldsymbol{P} \in \mathbb{R}^{n \times n}$. Then $\boldsymbol{A}$ must be a low-pass filter. For all $\boldsymbol{z} \in \mathbb{R}^n$, $\lim_{t \to \infty}\|\mathcal{HC}[\boldsymbol{A}^t\boldsymbol{z}]\|_2/\|\mathcal{DC}[\boldsymbol{A}^t\boldsymbol{z}]\|_2 = 0$.*

Theorem 1 is a straightforward result of Perron-Frobenius theorem. See Appendix B.1 for a proof. Theorem 1 also reveals that no matter how attention is computed inside the softmax function, including dot product (Vaswani et al., 2017), linear combination (Veličković et al., 2018), or L2 distance (Kim et al., 2021), the resulting attention matrix is always a low-pass filter. One can see consecutively applying self-attention matrix simulates the process of ViT's forward propagation. As the layer number increases infinitely, the final output will only keep the DC bias, and ViT loses all the feature expressive power.

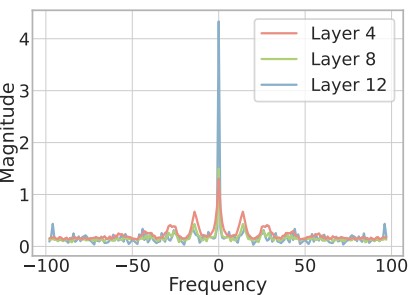

Figure 2: Visualize the spectral response of an attention map. We randomly pick a sample and depict its first head of 4/8/12-th layer. Refer to Appendix F.2.

**Corollary 2.** *Let $\boldsymbol{P}_1, \boldsymbol{P}_2, \cdots, \boldsymbol{P}_n$ be a sequence of matrix in $\mathbb{R}^{n \times n}$, and each $\boldsymbol{P}_k, \forall k = 1, \cdots, L$ has $\boldsymbol{A}_k = \mathrm{softmax}(\boldsymbol{P}_k)$. Then $\prod_{k=1}^{L} \boldsymbol{A}_k$ is also a low-pass filter.*

In fact, ViT re-computes self-attention matrices per layer, which seems to avoid the consecutive power of an identical self-attention matrix. However, we also provide Corollary 2, which suggests even the ViT consists of distinctive self-attention matrix at each layer, their composition turns out to act like a low-pass filter as well. We also visualize the spectrum of attention maps (Fig. 2 and more on Appendix F.2) to support our theoretical conclusions.

Knowing that self-attention matrices amount to low-pass filters, we are also interested in to which extent an MSA layer would suppress the high-frequency component. Thereby, we also provide a convergence rate to illustrate this speed that the high-frequency component are being annihilated.

**Theorem 3.** *(smoothening rate of SA) Let $\boldsymbol{A} = \mathrm{softmax}(\boldsymbol{P})$ and $\alpha = \max_{i,j}|\boldsymbol{P}_{ij}|$, where $\boldsymbol{P} \in \mathbb{R}^{n \times n}$. Define $\mathrm{SA}(\boldsymbol{X}) = \boldsymbol{A}\boldsymbol{X}\boldsymbol{W}_V$ as the output of a self-attention module, then*

$$\|\mathcal{HC}\left[\mathrm{SA}(\boldsymbol{X})\right]\|_F \leq \sqrt{\frac{ne^{2\alpha}}{e^{2\alpha} + n - 1}}\|\boldsymbol{W}_V\|_2 \|\mathcal{HC}\left[\boldsymbol{X}\right]\|_F. \tag{6}$$

*In particular, when $\boldsymbol{P} = \boldsymbol{X}\boldsymbol{W}_Q(\boldsymbol{X}\boldsymbol{W}_K)^T/\sqrt{d}$, and assume tokens are distributed inside a ball with radius $\gamma > 0$, i.e., $\|\boldsymbol{x}_i\|_2 \leq \gamma, \forall i = 1, \cdots, n$, then $\alpha \leq \gamma^2 \|\boldsymbol{W}_Q\boldsymbol{W}_K^T\|_2/\sqrt{d}$.*

The proof of Theorem 3 can be found in Appendix B.3. Theorem 3 says the high-frequency intensity ratio to the pre- and post- attention aggregation is upper bounded by $\|\boldsymbol{W}_V\|_2\sqrt{\frac{ne^{2\alpha}}{e^{2\alpha}+n-1}}$. When $\|\boldsymbol{W}_V\|_2\sqrt{\frac{ne^{2\alpha}}{e^{2\alpha}+n-1}} < 1$, $\mathcal{HC}\left[\mathrm{SA}(\boldsymbol{X})\right]$ converges to zero exponentially. We note that, no matter how attention is computed or signals are initialized, since $\sqrt{\frac{ne^{2\alpha}}{e^{2\alpha}+n-1}}$ is bounded by $\sqrt{n}$, $\|\boldsymbol{W}_V\|_2 < 1/\sqrt{n}$ will definitely cause a monotonically decreasing high-frequency component. When dot-product attention is adopted, a sufficient condition that $\|\mathcal{HC}\left[\boldsymbol{X}\right]\|_F$ decreases to zero within logarithmic time is $\gamma^2\|\boldsymbol{W}_Q\boldsymbol{W}_K^T\|_2/\sqrt{d} + \log\|\boldsymbol{W}_V\|_2 \leq \log\frac{n-1}{n}/2$.

### 2.3 Existing Mechanisms that Counteract Low-Pass Filtering

In this section, we take other ViT building blocks into consideration. We will justify whether Multi-Head Self-Attention (MSA), Feed-Forward Network (FFN), and residual connections can effectively alleviate the low-pass filtering drawbacks. All the derivations follow from Theorem 3, and some proof ideas are borrowed from Dong et al. (2021). We further present Fig. 1 to justify our results.

**Does multi-head help?** MSA employs weights to combine the results of multiple self-attention blocks. We can rewrite it as $\mathrm{MSA}(\boldsymbol{X}) = \sum_{h=1}^{H} \mathrm{SA}(\boldsymbol{X})\boldsymbol{W}_O^h$. We show by Proposition 4 in Appendix C.1 that the convergence rate turns to $\sigma_1\sigma_2 H\sqrt{\frac{ne^{2\alpha}}{e^{2\alpha}+n-1}}$, where $H$ is the number of heads, $\sigma_1 = \max_{h=1}^{H}\|\boldsymbol{W}_V^h\|_2$ and $\sigma_2 = \max_{h=1}^{H}\|\boldsymbol{W}_O^h\|_2$. One can see MSA can only slow down the convergence up to a constant $\sigma_2 H$, which does not root out the problem.

**Does residual connection benefit?** In addition to MSA, a transformer block also leverages a skip connection, which can be formulated as $\mathrm{Res}(\boldsymbol{X}) = \mathrm{MSA}(\boldsymbol{X}) + \boldsymbol{X}$. We show that residual connection

can effectively prevent high-frequency component from diminishing to zero by promoting the rate $\sigma_1\sigma_2 H\sqrt{\frac{ne^{2\alpha}}{e^{2\alpha}+n-1}}$ to $1 + \sigma_1\sigma_2 H\sqrt{\frac{ne^{2\alpha}}{e^{2\alpha}+n-1}} > 1$. Refer to Proposition 5 in Appendix C.2.

**Does FFN make any difference?** A feed-forward network is appended to MSA module. We characterize its effect in Appendix C.3. Our Proposition 6 suggests that a FFN with Lipschitz constant $\sigma_3$ contributes a $\sigma_3\left(1 + \sigma_1\sigma_2 H\sqrt{\frac{ne^{2\alpha}}{e^{2\alpha}+n-1}}\right)$ convergence rate, which does not improve the original one. However, if skip connection is adopted over FFN, $\sigma_3 > 1$ can guarantee the upper bound of the high-frequency component is non-contractive.

Although multi-head, FFN, and skip connection all help preserve the high-frequency signals, none would change the fact that MSA block as a whole only possesses the representational power of low-pass filters. Our Proposition 4, 5, 6 states multi-head, FFN, skip connections can only slow down the convergence by indistinguishably amplifying low- and high-frequency components with the same factor. However, since they are incapable of promoting high-frequency information separately, it is inevitable that high-frequency components are continuously diluted as ViT goes deeper. This restricts the expressiveness of ViT, resulting in the performance saturation in deeper ViT.

## 2.4 CONNECTIONS TO EXISTING THEORETIC UNDERSTANDING WORKS

It is known that Graph Convolutional Networks (GCN) are not more than low-pass filters (NT & Maehara, 2021). In the meanwhile, Oono & Suzuki (2019); Cai & Wang (2020) pointed out GCN's node features will be exponentially trapped into the nullspace of the graph Laplacian matrix. Similarly, our work concludes that self-attention module is yet another low-pass filter. Combining with our theoretical derivation, one can see the root reason is that both graph Laplacian matrices and self-attention matrices consistently own a fixed leading eigenvector, namely the DC basis. This makes aggregating information via such matrices inherently project the token representation onto these invariant eigenspaces. And we note that over-smoothing, rank collapse, and patch uniformity are all the manifestation of excessive low-pass filtering. See Appendix D.1 for more discussion.

In Dong et al. (2021), the authors proved that ViT's feature maps will doubly exponentially collapses to a rank-1 matrix, which reveals ViT loses feature expressiveness at deep layers. Besides, they also gave a systematic study on other building blocks of transformer. While they share the similar insights with us, our work further specifies which rank-1 matrix the feature activation will converge to, namely the subspace spanned by the DC basis. That makes our theory to be better grounded with signal-processing and geometric interpretations, via directly measuring the intensity of the high-frequency residual, instead of examining a composite norm distance to an agnostic rank-1 matrix. Although Dong et al. (2021) presented a faster convergence speed, we respectfully suggest that the current proof of Dong et al. (2021) might be deficient, or at least incomplete in the assumptions (see Appendix D.2). Moreover, our theory can be generalized to other attention mechanisms such as logistic attention (Veličković et al., 2018) and L2 distance (Kim et al., 2021). See Appendix D.3.

## 3 ATTNSCALE & FEATSCALE: SCALING FROM THE FOURIER-DOMAIN

### 3.1 ATTNSCALE: MAKE ATTENTION AN ALL-PASS FILTER

As we discussed in Section 2.2, self-attention matrix can only perform low-pass filtering, which narrows the filter space ViT can express. Inspired by this, we propose a scaling techniques directly manipulating the attention map, termed *Attention Scaling* (**AttnScale**), to balance the effects of low- and high-pass filtering and produce all-pass filters. AttnScale decomposes the self-attention matrix to a low-pass filter plus a high-pass filter, and introduces a trainable parameter to rescale the high-pass filter to match the magnitude with the low-pass component.

Formally, let $A$ denote a self-attention matrix. To decompose a low-pass filter from $A$, we find the *largest possible* low-pass filter that can be extracted from $A$. We use Lemma 8 in Appendix E to justify our solution. By Lemma 8, we can simply extract $L = \mathcal{F}^{-1}\operatorname{diag}(1,0,\cdots,0)\mathcal{F} = \mathbf{1}\mathbf{1}^T/n$ from $A$ and take the complementary part as the high-pass filter. Afterward, we can rescale the high-pass component of the filter, and combine low-pass and high-pass together to form a new self-attention matrix. We illustrate this scaling trick in Fig. 3b. To be more precise, for the $l$-th layer and $h$-th head, we recompute the self-attention map as follows:

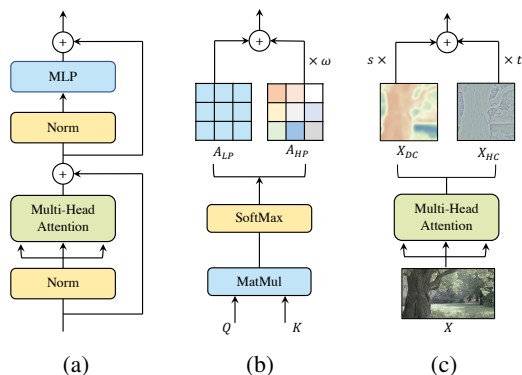

(a)        (b)        (c)

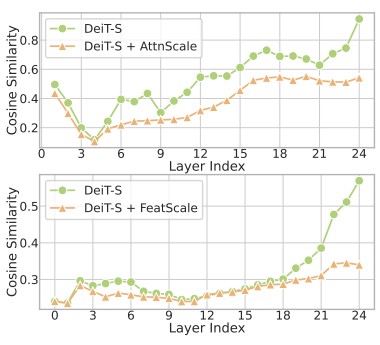

Figure 3: Illustration of our proposed techniques. **(a)** recalls the standard ViT block. **(b)** and **(c)** illustrate our proposed AttnScale and FeatScale, which scaling high-pass filter component and high-frequency signals, respectively.

Figure 4: Visualize cosine similarity of attention and feature maps with/without our proposed methods. Refer to Appendix F.3 for details.

$$\boldsymbol{A}_{LP}^{(l,h)} = \frac{1}{n}\mathbf{1}\mathbf{1}^T, \tag{7}$$

$$\boldsymbol{A}_{HP}^{(l,h)} = \boldsymbol{A}^{(l,h)} - \boldsymbol{A}_{LP}^{(l,h)}, \tag{8}$$

$$\hat{\boldsymbol{A}}^{(l,h)} = \boldsymbol{A}_{LP}^{(l,h)} + (\omega_{l,h} + 1)\boldsymbol{A}_{HP}^{(l,h)}, \tag{9}$$

where $\omega_{l,h}$ is a trainable parameter, and different layers and heads adopt separate $\omega_{l,h}$. During training time, $\omega_{l,h}$'s are initialized with 0, and jointly tuned with the other network parameters. By adjusting $\omega_{l,h}$, $\hat{\boldsymbol{A}}^{(l,h)}$ can simulate any type of filters: low-pass, high-pass, band-pass, or all-pass. Note that our AttnScale is extremely lightweight, as it only brings $O(HL)$ extra parameters, where $H$ is the number of heads, and $L$ is the number of ViT blocks.

## 3.2 FEATSCALE: REWEIGHT HIGH-FREQUENCY SIGNALS

According to our analysis in Section 2.2, MSA module will indiscriminately suppress high-frequency signals, which leads to severe information loss. Even though residual connection can retrieve lost information through the skip path, the high-frequency portion will be inevitably diluted (Theorem 1). To this end, we propose another scaling technique that operates on feature maps, named *Feature Scaling* (**FeatScale**). FeatScale processes the output of MSA by mixing the information from varying frequency bands discriminatively. FeatScale first decomposes the resultant signals into their DC and high-frequency components. Then it introduces two groups of parameters to re-weight the two components for each channel, respectively. The pipeline of this scaling technique is depicted in Fig. 3c. To be more precise, we re-weight the output of the $l$-th MSA by

$$\boldsymbol{X}_{DC}^{(l)} = \mathcal{DC}\left[\mathrm{MSA}(\boldsymbol{X})\right](\mathrm{diag}(\boldsymbol{s}_l) + \boldsymbol{I}), \tag{10}$$

$$\boldsymbol{X}_{HC}^{(l)} = \mathcal{HC}\left[\mathrm{MSA}(\boldsymbol{X})\right](\mathrm{diag}(\boldsymbol{t}_l) + \boldsymbol{I}), \tag{11}$$

$$\boldsymbol{X}^{(l)} = \boldsymbol{X}_{DC}^{(l)} + \boldsymbol{X}_{HC}^{(l)}, \tag{12}$$

where $\boldsymbol{s}_l \in \mathbb{R}^d$ and $\boldsymbol{t}_l \in \mathbb{R}^d$ are learnable parameters to perform channel-wise re-weighting. We initialize $\boldsymbol{s}_l$ and $\boldsymbol{t}_l$ with zeros and tune them with gradient descent. After adjusting the proportion of different frequency signals, FeatScale can prevent the dominance of the DC component. $\mathcal{DC}\left[\cdot\right]$ and $\mathcal{HC}\left[\cdot\right]$ are cheap to compute without explicit Fourier transform. Calculating $\mathcal{DC}\left[\boldsymbol{X}\right]$ is as simple as running the column average of matrix $\boldsymbol{X}$, and $\mathcal{HC}\left[\boldsymbol{X}\right]$ can be efficiently computed by $\boldsymbol{X} - \mathcal{DC}\left[\boldsymbol{X}\right]$.

## 3.3 DISCUSSION

We have proposed two methods to facilitate the deeper stacking of ViT MSA modules, and discussed their motivations and strengths from our perspective of filtering and signal processing. In this section, we will connect these two techniques with commonly mentioned problems with ViTs.

**How does AttnScale prevent attention collapse?** Deep ViT suffers from the attention collapse issue (Zhou et al., 2021a). When transformer goes deepr, the attention maps gradually become similar

and even much the same after certain layers. Combining with our thoery, one can see collapsed attention maps turn out to be a pure low-pass filter, which wipes off all the high-frequency information in one shot. Zhou et al. (2021a) proposed the re-attention trick, which blends attention map across different heads. By doing this, modified attention maps aggregate high-pass components from other heads and is endowed with richer filtering property. We note that our AttnScale is akin to a more lightweight re-attention mechanism with better interpretability. We rewrite the attention map as a sum of an already-collapsed attention ($\mathbf{1}\mathbf{1}^T/n$) with the complementary residual map that encodes diverse patterns in a self-attention matrix. By re-weighting the residual map, the diversified patterns can be amplified, which prevents it from degenerating to a rank-1 matrix. We further verify this argument using cosine similarity metric (Zhou et al., 2021a) in the upper sub-figure of Fig. 4.

**How does FeatScale conserve patch diversity?**   Self-attention blocks tend to map different patches into similar latent representations, yielding information loss and performance degradation (Gong et al., 2021). By our theory, this asymptotic smoothness of feature map is caused by excessive low-pass filtering, and the remaining DC bias signifies uniform patch representations. Conventional approaches to addressing this problem include incorporating convolutional layers (Wu et al., 2021; Jiang et al., 2021b) and enforcing patch diversity regularizations (Gong et al., 2021). As diverse features are often characterized by high-frequency signals, our FeatScale instead elevating the high-frequency component via a learnable scaling factor, can be regarded as a more straightforward way to reconstruct the patch richness. Compared with LayerScale (Touvron et al., 2021b), in which each frequency band is equally scaled, our FeatScale treating DC and high-frequency components differently, not only perform a per-channel normalization, but also perform a spectral-domain calibration with high-frequency details and low-frequency characteristics. The lower sub-plot of Fig. 4 shows ViT with our FeatScale has lower feature similarity at deep layer.

# 4    RELATED WORK

**Transformers in Vision.**   Transformer (Vaswani et al., 2017) entirely relies on self-attention mechanism to capture correlation and exchange information globally among the input. It has achieved a remarkable performance in natural language processing (Devlin et al., 2018; Dai et al., 2019; Brown et al., 2020) and many cross-disciplinary applications (Jumper et al., 2021; Ying et al., 2021; Zheng et al., 2021b). Recent advances have also successfully applied Transformer to computer vision tasks. Dosovitskiy et al. (2020) first adopts a pure transformer architecture (ViT) for image classification. The follow-up works (Chen et al., 2021b) extend ViT to various vision tasks, such as object detection (Carion et al., 2020; Zhu et al., 2021; Zheng et al., 2021a; Sun et al., 2020), segmentation (Chen et al., 2021a; Wang et al., 2021), image generation (Parmar et al., 2018; Jiang et al., 2021a), video processing (Zhou et al., 2018; Arnab et al., 2021), and 3D instance processing (Guo et al., 2021; Lin et al., 2021). To capture multi-scale non-local contexts, Zhang et al. (2020) designs transformers in self-level, top-down, and bottom-up interaction fashion. Liu et al. (2021) presents hierarchical ViTs with shifted window based attention that can efficiently extract multi-scale features. To dismiss ViT from the heavy reliance on large-scale dataset pre-training, Touvron et al. (2021a); Yuan et al. (2021) propose knowledge distillation and progressive tokenization for data-efficient training. Despite impressive effectiveness, most of these model are only based on relatively shallow ViT backbones with a dozen of MSA blocks.

**Advances in deep ViTs.**   Building deeper ViTs has arisen many interests. Zhou et al. (2021a) first investigated the depth scalability of ViT. The authors found that the attention collapse hinders ViT from scaling up, and propose two methods to conquer this problem i) increasing the embedding dimension, and ii) a cross-head re-attention trick to regenerate attention map. A concurrent work Touvron et al. (2021b) came up with a LayerScale layer that performs per-channel multiplication for each residual block. More importantly, they make explicit separation of transformer layers involving self-attention between patches, from class-attention layers that are devoted to extract the global content into a single embedding to be decoded. Gong et al. (2021) further proposed a series of losses that can enforce patch diversity in ViT. Such regularizations include penalty on cosine similarity, patch-wise contrastive loss, and mixing loss. Tang et al. (2021) presented a shortcut augmentation scheme with block-circulant projection to improve feature diversity. Although these existing solutions manage to deepen ViTs, most of them are empirical works and bring no principled theory.

**Role of depth in NNs.**   Discussing the importance of deep structures in Neural Networks (NNs) is an overly broad topic. Here we only focus on a subset of works that scaling up a transformer could

Table 1: Experimental evaluation of AttnScale & FeatScale plugged into DeiT and CaiT. The number inside the (↑ ·) represents the performance gain compared with the baseline model, and accuracies within/out of parenthesis are the reported/reproduced performance.

| Backbone | Method | Input size | # Layer | # Param | FLOPs | Throughput | Top-1 Acc (%) |
|---|---|---|---|---|---|---|---|
| DeiT | DeiT-S | 224 | 12 | 22.0M | 4.57G | 1589.4 | 79.8 (79.9) |
| | DeiT-S + AttnScale | 224 | 12 | 22.0M | 4.57G | 1416.7 | 80.7 (↑ 0.9) |
| | DeiT-S + FeatScale | 224 | 12 | 22.0M | 4.57G | 1509.9 | 80.9 (↑ 1.1) |
| | DeiT-S | 224 | 24 | 43.3M | 9.09G | 836.4 | 80.5 (81.0) |
| | DeiT-S + AttnScale | 224 | 24 | 43.3M | 9.10G | 722.0 | 81.1 (↑ 0.6) |
| | DeiT-S + FeatScale | 224 | 24 | 43.4M | 9.10G | 772.5 | 81.3 (↑ 0.8) |
| CaiT | CaiT-S | 224 | 24 | 46.9M | 9.33G | 371.9 | 82.6 (82.7) |
| | CaiT-S + AttnScale | 224 | 24 | 46.9M | 9.34G | 339.0 | 83.2 (↑ 0.6) |
| | CaiT-S + FeatScale | 224 | 24 | 46.9M | 9.34G | 358.2 | 83.2 (↑ 0.6) |
| Swin | Swin-S | 224 | 24 | 49.6M | 8.74G | 593.2 | 83.0 (83.0) |
| | Swin-S + AttnScale | 224 | 24 | 49.6M | 8.75G | 553.4 | 83.4 (↑ 0.4) |
| | Swin-S + FeatScale | 224 | 24 | 49.6M | 8.75G | 550.3 | 83.5 (↑ 0.5) |

relate to. For ordinal deep learning models, such as FFNs and CNNs, deep architecture immediately benefits from the universal approximation power and expressive capacity (Cybenko, 1989; Hornik, 1991; Telgarsky, 2016; Lu et al., 2017; Petersen & Voigtlaender, 2020; Zhou, 2020). In contrast, several studies in graph learning domain have reported severe performance degradation due to over-smoothing when stacking many layers (Kipf & Welling, 2017; Wu et al., 2019; Li et al., 2018). The subsequent studies (NT & Maehara, 2021; Oono & Suzuki, 2019; Cai & Wang, 2020) gave theoretical explanations of the over-smoothing phenomena from the views of graph signal filtering and feature dynamics. Likewise, ViT have been witnessed performance saturation when going deeper. However, to our best knowledge, Dong et al. (2021) is the sole work in the literature that systematically and rigorously analyzes this issue with deep ViT. The main idea of this work is showing the self-attention block will downgrade the rank of the feature maps. Our work takes one step forward by studying ViT on spectral domain, and manages to reveal the signals will ultimately fall into the one-dimension DC subspace. We see our theory and techniques are also applicable to NLP transformers. However, we only focus on ViT because empirical observations indicate NLP modeling (including Transformer) does not require a deep structure (Vaswani et al., 2017; Brown et al., 2020), while vision tasks always demand one (LeCun et al., 2015).

## 5 EXPERIMENTS

In this section, we report experiment results to validate our proposed methods. First, we validate the effectiveness of our AttnScale and FeatScale when integrated with different deep ViT backbones (Section 5.1). Second, we compare our best models with state-of-the-art (SOTA) results (Section 5.2). All of our experiments are conducted on the ImageNet dataset (Russakovsky et al., 2015) with around 1.3M images in the training set and 50k images in the validation set. Our implementations are based on Timm (Wightman, 2019) and DeiT (Touvron et al., 2021a) repositories.

### 5.1 HOW CAN ATTNSCALE & FEATSCALE BENEFIT DEEP VIT?

**Experiment Settings.** In this subsection, we intend to testify our models are beneficial to various ViT backbones with different depth settings and training modes. We choose DeiT (Touvron et al., 2021a) as our first backbone in order to train from scratch. When training 12-layer DeiT, we follow the same training recipe, hyper-parameters, and data augmentation with Touvron et al. (2021a). When training 24-layer DeiT, we follow the setting in Gong et al. (2021). Specially, we set dropout rate to 0.2 when training 24-layer DeiT (Touvron et al., 2021b). Our second backbone is CaiT (Touvron et al., 2021b). We only apply our techniques to the patch embedding layers. The third backbone is the SOTA model Swin-Transformer (Liu et al., 2021). All experimental settings share the same with Liu et al. (2021). In addition to training from scratch, we also investigate the fine-tuning setting. We defer this part to Appendix G.1.

**Results.** All of our experimental evaluations are summarized in Table 1. The results suggest our proposed AttnScale and FeatScale successfully facilitate both DeiT, CaiT, Swin-Transformer under different depth settings and training modes. Specifically, AttnScale brings *less than 100/150 extra parameters for 12/24-layer DeiT* while boosting the top-1 accuracy by 0.9% for 12-layer DeiT and 0.6% for 24-layer DeiT. Our FeatScale substantially improves top-1 accuracy by 1% for 12-layer DeiT and 0.8% for 24-layer DeiT. Compared with existing techniques, the improvements

Table 2: Compared with state-of-the-art models on ImageNet dataset. Accuracies with superscript (∗) are reported by Gong et al. (2021), with superscript (†) are reported by Yuan et al. (2021), and others are reported by the original papers. Bold accuracies signifies best models among pure transformers.

| Category | Method | # Param | Input size | # Layer | Top-1 Acc (%) |
|---|---|---|---|---|---|
| CNN | ResNet-152 (He et al., 2016) | 230M | 224 | 152 | 78.1 * |
| | DenseNet-201 (Huang et al., 2017) | 77M | 224 | 201 | 77.6 * |
| CNN+ Transformer | CVT-21 (Wu et al., 2021) | 32M | 224 | 21 | 82.5 * |
| | LV-ViT-S (Jiang et al., 2021b) | 26M | 224 | 16 | 83.3 * |
| Transformer | ViT-S/16 (Dosovitskiy et al., 2020) | 49M | 224 | 12 | 78.1 † |
| | ViT-B/16 (Dosovitskiy et al., 2020) | 86M | 224 | 12 | 79.8 † |
| | DeiT-S (Touvron et al., 2021a) | 22M | 224 | 12 | 79.8 |
| | DeiT-S Distilled (Touvron et al., 2021a) | 22M | 224 | 12 | 81.2 |
| | Swin-S (Liu et al., 2021) | 50M | 224 | 12 | 83.0 |
| | T2T-ViT-24 (Yuan et al., 2021) | 64M | 224 | 24 | 82.3 |
| | DeepViT-24B (Zhou et al., 2021a) | 36M | 224 | 24 | 80.1 |
| | CaiT-S (Touvron et al., 2021b) | 47M | 224 | 24 | 82.7 |
| | DeiT-S + DiversePatch (Gong et al., 2021) | 44M | 224 | 24 | 82.2 |
| Ours | DeiT-S + AttnScale | 43M | 224 | 24 | 81.1 |
| | DeiT-S + FeatScale | 43M | 224 | 24 | 81.3 |
| | CaiT-S + AttnScale | 47M | 224 | 24 | **83.2** |
| | CaiT-S + FeatScale | 47M | 224 | 24 | **83.2** |
| | Swin-S + AttnScale | 50M | 224 | 24 | **83.4** |
| | Swin-S + FeatScale | 50M | 224 | 24 | **83.5** |

of AttnScale and FeatScale already surpass re-attention (0.6%) (Zhou et al., 2021a), LayerScale (0.7%) (Touvron et al., 2021b), and late class token insertion (0.6%) (Touvron et al., 2021b). We also observe a consistent 0.6% performance gain when AttnScale and FeatScale plugged into CaiT. Under fine-tuning setting, as we will show in Appendix G.1, only *tens of epoch's fine-tuning* can further promote their performance by $\geq 0.2\%$ (see Table 3). On Swin-Transformer, both our AttnScale and FeatScale bring around 0.5% accuracy gain. This makes Swin-S with 50M parameters even comparable to Swin-B (83.5% top-1 accuracy on ImageNet1k) with 88M parameters. We defer more model interpretation and visualization to Appendix G. For a brief summary, we observe that both shallow and deep ViT enjoy from AttnScale and FeatScale that: 1) the attention maps can simulate richer filtering properties (compare Fig. 9 with Fig. 10), and 2) more high-frequency data can be preserved (refer to Fig. 11).

## 5.2 COMPARISON WITH SOTA MODELS

In this subsection, we compare our best models with state-of-the-art models on ImageNet benchmark. We choose SOTA models from three classes: CNN only, CNN + transformer, and pure transformer. For transformer domain, we only conduct experiments with those lightweight models with comparable number of parameters, such as ViT-S and DeiT-S. All the results are presented in Table 2.

Among all methods, Swin-Transformer combined with our methods achieves the state-of-the-art performance. Our CaiT-S + AttnScale and CaiT-S + FeatScale on 24-layer CaiT-S also attain superior results over all other pure transformers, while keeping low parameter cost. That our performance surpasses some CNN-based models (e.g., ResNet-152 and CVT) indicates by increasing depth, ViT will be endowed with higher potential to surpass CNNs that have been dominating computer vision domain so far. Our DeiT-S+FeatScale result also outperforms ViT-B/16 and DeiT-S Distilled, which suggests deepening network can bring more considerable accuracy gain than increasing model width or employing a teacher model.

## 6 CONCLUSION

In this paper, we investigate the scalability issue with ViT and propose two practical solutions via Fourier domain analysis. Our theoretical findings indicate Multi-Head Self-Attention (MSA) inherently performs low-pass filtering on image signals, thus causes rank collapse and patch uniformity problems in deep ViT. To this end, we proposed two techniques, AttnScale and FeatScale, that can effectively break such low-pass filtering bottleneck by adaptively scaling high-pass filter component and high-frequency signals, respectively. Our experiments also validate the effectiveness of our methods. Both techniques can boost various ViT backbones by a significant performance gain. Grounded with our theoretical framework, interesting directions for further work include designing parameter regularizations and spectrum-specific normalization layers.

ACKNOWLEDGMENTS

Z.W. is in part supported by an NSF SCALE MoDL project (#2133861).

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

## A  MORE PRELIMINARIES ON FOURIER ANALYSIS

In this appendix, we provide more preliminary knowledge about Fourier analysis. Here, we only consider discrete Fourier transform on real-value domain $\mathcal{F} : \mathbb{R}^n \to \mathbb{C}^n$. A Discrete Fourier transform (DFT) can be written in a matrix form as below [1]:

$$\boldsymbol{DFT} = \frac{1}{\sqrt{n}} \begin{bmatrix} 1 & 1 & \cdots & 1 \\ 1 & e^{2\pi\mathrm{j}} & \cdots & e^{2\pi\mathrm{j}(n-1)} \\ \vdots & \vdots & \ddots & \vdots \\ 1 & e^{2\pi\mathrm{j}(k-1)\cdot 1} & \cdots & e^{2\pi\mathrm{j}(k-1)\cdot(n-1)} \\ \vdots & \vdots & \ddots & \vdots \\ 1 & e^{2\pi\mathrm{j}(n-1)} & \cdots & e^{2\pi\mathrm{j}(n-1)^2} \end{bmatrix}, \tag{13}$$

and its inverse discrete Fourier transform is $\boldsymbol{DFT}^{-1} = \overline{\boldsymbol{DFT}}$. In this paper, we regard matrices as multi-channel signals. For example, $\boldsymbol{X} \in \mathbb{R}^{n \times d}$ means $d$-channel $n$-length signals. When DFT and inverse DFT are applies to multi-channel signals, each channel is transformed independently, i.e., $\mathcal{F}(\boldsymbol{X}) = [\mathcal{F}(\boldsymbol{x}_1) \quad \cdots \quad \mathcal{F}(\boldsymbol{x}_d)] = \boldsymbol{DFT} \cdot \boldsymbol{X}$.

Hereby, we can simply operators $\mathcal{DC}\left[\cdot\right]$ and $\mathcal{HC}\left[\cdot\right]$ using the matrices in Eqn. 13. By definition, we can write $\mathcal{DC}\left[\cdot\right]$ as below:

$$\mathcal{DC}\left[\boldsymbol{x}\right] = \boldsymbol{DFT}^{-1} \operatorname{diag}(1, 0, \cdots, 0) \boldsymbol{DFT} \boldsymbol{x} \tag{14}$$

$$= \frac{1}{n} \mathbf{1}\mathbf{1}^T \boldsymbol{x}, \tag{15}$$

namely $\mathcal{DC}\left[\cdot\right] = \mathbf{1}\mathbf{1}^T / n$. Conversely, we can write $\mathcal{HC}\left[\cdot\right]$ as:

$$\mathcal{HC}\left[\boldsymbol{x}\right] = \boldsymbol{DFT}^{-1} \operatorname{diag}(0, 1, \cdots, 1) \boldsymbol{DFT} \boldsymbol{x} \tag{16}$$

$$= \boldsymbol{DFT}^{-1} (\boldsymbol{I} - \operatorname{diag}(1, 0, \cdots, 0)) \boldsymbol{DFT} \boldsymbol{x} \tag{17}$$

$$= \boldsymbol{I} - \frac{1}{n} \mathbf{1}\mathbf{1}^T \boldsymbol{x}, \tag{18}$$

which indicates $\mathcal{HC}\left[\cdot\right] = \boldsymbol{I} - \mathbf{1}\mathbf{1}^T / n$. We will frequently use these derivations later in the proofs.

---

[1]Without loss of generality, we can only consider 1D Fourier transformer, since the DC components are invariant to the dimension of signals.

# B  DEFERRED PROOFS

## B.1  PROOF OF THEOREM 1

**Theorem 1.** *(SA matrix is a low-pass filter) Let $\boldsymbol{A} = \text{softmax}(\boldsymbol{P})$, where $\boldsymbol{P} \in \mathbb{R}^{n \times n}$. Then $\boldsymbol{A}$ must be a low-pass filter. For all $\boldsymbol{z} \in \mathbb{R}^n$,*

$$\lim_{t \to \infty} \frac{\|\mathcal{HC}\left[\boldsymbol{A}^t \boldsymbol{z}\right]\|_2}{\|\mathcal{DC}\left[\boldsymbol{A}^t \boldsymbol{z}\right]\|_2} = 0.$$

*Proof.* Let $\lambda_1, \lambda_2, \cdots, \lambda_s \in \mathbb{C}$ be the eigenvalues of $\boldsymbol{A}$ with ordering $|\lambda_1| \geq |\lambda_2| \geq \cdots \geq |\lambda_s|$. Notice that, $\boldsymbol{A}$ is a positive matrix, each of whose element is strictly greater than zero ($\boldsymbol{A}_{ij} > 0, \forall i, j$). Besides, for all $i = 1, \cdots, n$, $\sum_{j=1}^{n} \boldsymbol{A}_{i,j} = 1$. Therefore, $\boldsymbol{A}\mathbf{1} = \mathbf{1}$ implies $\boldsymbol{A}$ must have an eigenvalue 1 and its corresponding eigenvector is the all-one vector $\mathbf{1}$.

By Perron-Frobenius Theorem (Meyer, 2000), eigenvalue 1 corresponds to a all-positive eigenvector $\mathbf{1}$, implies $\lambda_1 = 1$ should be the largest eigenvalue without multiplicity, and the absolute value of other eigenvalues $\lambda_2, \cdots, \lambda_s$ must be less than 1. Let us rewrite $\boldsymbol{A}$ in the Jordan canonical form $\boldsymbol{A} = \boldsymbol{P}\boldsymbol{J}\boldsymbol{P}^{-1}$:

$$\boldsymbol{A} = \underbrace{\begin{bmatrix} \boldsymbol{v}_1 & \cdots & \boldsymbol{v}_n \end{bmatrix}}_{\boldsymbol{P}} \underbrace{\begin{bmatrix} \lambda_1 & & & \\ & \boldsymbol{J}(\lambda_2) & & \\ & & \ddots & \\ & & & \boldsymbol{J}(\lambda_s) \end{bmatrix}}_{\boldsymbol{J}} \underbrace{\begin{bmatrix} \boldsymbol{u}_1^T \\ \vdots \\ \boldsymbol{u}_n^T \end{bmatrix}}_{\boldsymbol{P}^{-1}}, \tag{19}$$

where the Jordan block $\boldsymbol{J}(\lambda)$ can be written as

$$\boldsymbol{J}(\lambda) = \begin{bmatrix} \lambda & 1 & & \\ & \lambda & 1 & \\ & & \ddots & \ddots & \\ & & & \lambda & 1 \\ & & & & \lambda \end{bmatrix}. \tag{20}$$

Applying $\boldsymbol{A}$ to $\boldsymbol{z}$ for $t$ times can be written as $\boldsymbol{A}^t \boldsymbol{z}$ which is equivalent to:

$$\boldsymbol{A}^t \boldsymbol{z} = \boldsymbol{P}\boldsymbol{J}^t \boldsymbol{P}^{-1} \boldsymbol{z} = \boldsymbol{P} \begin{bmatrix} \lambda_1 & & & \\ & \boldsymbol{J}(\lambda_2) & & \\ & & \ddots & \\ & & & \boldsymbol{J}(\lambda_s) \end{bmatrix}^t \boldsymbol{P}^{-1} \boldsymbol{z} \tag{21}$$

$$= \boldsymbol{P} \begin{bmatrix} \lambda_1^t & & & \\ & \boldsymbol{J}(\lambda_2)^t & & \\ & & \ddots & \\ & & & \boldsymbol{J}(\lambda_s)^t \end{bmatrix} \boldsymbol{P}^{-1} \boldsymbol{z} \tag{22}$$

Let $f(x) = x^t$, then $\boldsymbol{A}^t = \boldsymbol{P}f(\boldsymbol{J})\boldsymbol{P}^{-1} = \boldsymbol{P}\,\text{diag}(f(\lambda_1), f(\boldsymbol{J}(\lambda_2)), \cdots, f(\boldsymbol{J}(\lambda_s)))\boldsymbol{P}^{-1}$. Suppose a Jordan block with shape $k \times k$, then

$$f(\boldsymbol{J}(\lambda)) = \begin{bmatrix} f(\lambda) & f'(\lambda) & \frac{f''(\lambda)}{2!} & \cdots & \frac{f^{(k-1)}(\lambda)}{(k-1)!} \\ & f(\lambda) & f'(\lambda) & \ddots & \vdots \\ & & \ddots & \ddots & \frac{f''(\lambda)}{2!} \\ & & & f(\lambda) & f'(\lambda) \\ & & & & f(\lambda) \end{bmatrix} \tag{23}$$

Therefore, on the diagonal number $m \leq \min(t, k-1)$ above the main diagonal stands:

$$\frac{t(t-1)...(t-m+1)}{m!} \lambda^{t-m} \tag{24}$$

For arbitrary $m \le k - 1$ and $|\lambda| < 1$,

$$\lim_{t \to \infty} \frac{t(t-1)...(t-m+1)}{m!} \lambda^{t-m} = 0 \tag{25}$$

Recall that $\lambda_1 = 1$ and according to the definition of Jordan canonical form, $\boldsymbol{v}_1 = \boldsymbol{1}$. By Eqn. 25:

$$\lim_{t \to \infty} \boldsymbol{A}^t \boldsymbol{z} = \boldsymbol{P} \lim_{t \to \infty} \mathrm{diag}(f(\lambda_1), f(\boldsymbol{J}(\lambda_2)), \cdots, f(\boldsymbol{J}(\lambda_s))) \boldsymbol{P}^{-1} \boldsymbol{z} \tag{26}$$

$$= \boldsymbol{P} \, \mathrm{diag}(\lambda_1^t, \boldsymbol{0}, \cdots, \boldsymbol{0}) \boldsymbol{P}^{-1} \boldsymbol{z} \tag{27}$$

$$= \lambda_1^t \boldsymbol{v}_1 \boldsymbol{u}_1^T \boldsymbol{z} \tag{28}$$

$$= \boldsymbol{1} \boldsymbol{u}_1^T \boldsymbol{z} \tag{29}$$

Plug the result from Eqn. 29 into the original limit:

$$\lim_{t \to \infty} \frac{\|\mathcal{HC}\,[\boldsymbol{A}^t \boldsymbol{z}]\|_2}{\|\mathcal{DC}\,[\boldsymbol{A}^t \boldsymbol{z}]\|_2} = \lim_{t \to \infty} \sqrt{\frac{\|\mathcal{HC}\,[\boldsymbol{A}^t \boldsymbol{z}]\|_2^2}{\|\boldsymbol{z} - \mathcal{HC}\,[\boldsymbol{A}^t \boldsymbol{z}]\|_2^2}} \tag{30}$$

$$= \lim_{t \to \infty} \sqrt{\frac{\|\mathcal{HC}\,[\boldsymbol{A}^t \boldsymbol{z}]\|_2^2}{\|\boldsymbol{z}\|_2^2 - \|\mathcal{HC}\,[\boldsymbol{A}^t \boldsymbol{z}]\|_2^2}} \tag{31}$$

$$= \lim_{t \to \infty} \sqrt{\frac{\|(\boldsymbol{I} - \frac{1}{n}\boldsymbol{1}\boldsymbol{1}^T)\boldsymbol{A}^t \boldsymbol{z}\|_2^2}{\|\boldsymbol{z}\|_2^2 - \|(\boldsymbol{I} - \frac{1}{n}\boldsymbol{1}\boldsymbol{1}^T)\boldsymbol{A}^t \boldsymbol{z}\|_2^2}} \tag{32}$$

$$= \sqrt{\frac{\|(\boldsymbol{I} - \frac{1}{n}\boldsymbol{1}\boldsymbol{1}^T)\boldsymbol{1}\boldsymbol{u}_1^T \boldsymbol{z}\|_2^2}{\|\boldsymbol{z}\|_2^2 - \|(\boldsymbol{I} - \frac{1}{n}\boldsymbol{1}\boldsymbol{1}^T)\boldsymbol{1}\boldsymbol{u}_1^T \boldsymbol{z}\|_2^2}} \tag{33}$$

$$= \sqrt{\frac{\|(\boldsymbol{I}\boldsymbol{1}\boldsymbol{u}_1^T \boldsymbol{z} - \boldsymbol{1}\boldsymbol{u}_1^T \boldsymbol{z}\|_2^2}{\|\boldsymbol{z}\|_2^2 - \|(\boldsymbol{I}\boldsymbol{1}\boldsymbol{u}_1^T \boldsymbol{z} - \boldsymbol{1}\boldsymbol{u}_1^T \boldsymbol{z}\|_2^2}} \tag{34}$$

$$= 0 \tag{35}$$

where Eqn. 31 is due to the orthogonality of DC and HC terms. $\square$

## B.2 Proof of Corollary 2

**Corollary 2.** *Let $\boldsymbol{P}_1, \boldsymbol{P}_2, \cdots, \boldsymbol{P}_n$ be a sequence of matrix in $\mathbb{R}^{n \times n}$, and each $\boldsymbol{P}_k, \forall k = 1, \cdots, L$ has $\boldsymbol{A}_k = \mathrm{softmax}(\boldsymbol{P}_k)$. Then $\prod_{k=1}^{L} \boldsymbol{A}_k$ is also a low-pass filter.*

*Proof.* Let $\boldsymbol{A} = \prod_{k=1}^{L} \boldsymbol{A}_k$, then we show $\boldsymbol{A}$ satisfies the following conditions, so that $\boldsymbol{A}$ can be regarded as another self-attention matrix. Then we can conclude the proof by Theorem 1.

1) For every $i = 1, \cdots, n$, $\sum_{j=1}^{n} \boldsymbol{A}_{ij} = 1$.

Suppose $\sum_{j=1}^{n} \boldsymbol{B}_{ij} = 1$ for every $i = 1, \cdots, n$, then for every $k = 1, \cdots, L$ and $i = 1, \cdots, n$,

$$\sum_{j=1}^{n} (\boldsymbol{A}_k \boldsymbol{B})_{ij} = \sum_{j=1}^{n} \sum_{m=1}^{n} \boldsymbol{A}_{k,im} \boldsymbol{B}_{mj} \tag{36}$$

$$= \sum_{m=1}^{n} \left( \boldsymbol{A}_{k,im} \left( \sum_{j=1}^{n} \boldsymbol{B}_{mj} \right) \right) \tag{37}$$

$$= \sum_{m=1}^{n} \boldsymbol{A}_{k,im} = 1. \tag{38}$$

By induction, for every $i$, $\sum_{j=1}^{n} \boldsymbol{A}_{1,ij} = 1 \Rightarrow \sum_{j=1}^{n} (\boldsymbol{A}_2 \boldsymbol{A}_1)_{ij} = 1 \Rightarrow \cdots \Rightarrow \sum_{j=1}^{n} \boldsymbol{A}_{ij} = 1$.

2) For every $i, j = 1, \cdots, n$, $\boldsymbol{A}_{ij} > 0$.

Suppose $\boldsymbol{B}_{ij} > 0, \forall i, j$, then for every $k = 1, \cdots, L$, $(\boldsymbol{A}_k \boldsymbol{B})_{ij} = \sum_{k=1}^{n} \boldsymbol{A}_{k,im} \boldsymbol{B}_{mj}$. Since $\boldsymbol{A}_{k,im} > 0, \boldsymbol{B}_{mj} > 0$, then $(\boldsymbol{A}_k \boldsymbol{B})_{ij} > 0$. By induction, for every $i, j$, $\boldsymbol{A}_{1,ij} > 0 \Rightarrow (\boldsymbol{A}_2 \boldsymbol{A}_1)_{ij} > 0 \Rightarrow \cdots \Rightarrow \boldsymbol{A}_{ij} > 0$. $\square$

### B.3 PROOF OF THEOREM 3

**Theorem 3.** *(convergence rate of SA) Let $\boldsymbol{A} = \mathrm{softmax}(\boldsymbol{P})$ and $\alpha = \max_{i,j} |\boldsymbol{P}_{ij}|$, where $\boldsymbol{P} \in \mathbb{R}^{n \times n}$. Define $\mathrm{SA}(\boldsymbol{X}) = \boldsymbol{A} \boldsymbol{X} \boldsymbol{W}_V$ as the output of a self-attention module, then*

$$\|\mathcal{HC}\left[\mathrm{SA}(\boldsymbol{X})\right]\|_F \leq \sqrt{\frac{ne^{2\alpha}}{e^{2\alpha} + n - 1}} \|\boldsymbol{W}_V\|_2 \|\mathcal{HC}\left[\boldsymbol{X}\right]\|_F.$$

*In particular, when $\boldsymbol{P} = \boldsymbol{X} \boldsymbol{W}_Q (\boldsymbol{X} \boldsymbol{W}_K)^T / \sqrt{d}$, and assume tokens are distributed inside a ball with radius $\gamma > 0$, i.e., $\|\boldsymbol{x}_i\|_2 \leq \gamma, \forall i = 1, \cdots, n$, then $\alpha \leq \gamma^2 \|\boldsymbol{W}_Q \boldsymbol{W}_K^T\|_2 / \sqrt{d}$.*

*Proof.* First, we write $\boldsymbol{X} = \mathcal{DC}\left[\boldsymbol{X}\right] + \mathcal{HC}\left[\boldsymbol{X}\right] = \boldsymbol{1}^T \boldsymbol{z} + \boldsymbol{H}$, where $\mathcal{DC}\left[\boldsymbol{X}\right] = \boldsymbol{1} \boldsymbol{z}^T$ equals to the orthogonal projection of $\boldsymbol{X}$ to subspace $\mathrm{span}(\boldsymbol{1})$, and $\boldsymbol{H} = \mathcal{HC}\left[\boldsymbol{X}\right]$ represents the remaining part of the original signals.

$$\mathcal{HC}\left[\mathrm{SA}(\boldsymbol{X})\right] = (\boldsymbol{I} - \boldsymbol{11}^T) \boldsymbol{A} \boldsymbol{X} \boldsymbol{W}_V \tag{39}$$

$$= \left(\boldsymbol{I} - \frac{1}{n} \boldsymbol{11}^T\right) \boldsymbol{A} (\boldsymbol{1} \boldsymbol{z}^T + \boldsymbol{H}) \boldsymbol{W}_V \tag{40}$$

$$= \left(\boldsymbol{I} - \frac{1}{n} \boldsymbol{11}^T\right) \boldsymbol{A} \boldsymbol{1} \boldsymbol{z}^T \boldsymbol{W}_V + \left(\boldsymbol{I} - \frac{1}{n} \boldsymbol{11}^T\right) \boldsymbol{A} \boldsymbol{H} \boldsymbol{W}_V \tag{41}$$

$$= \left(\boldsymbol{I} - \frac{1}{n} \boldsymbol{11}^T\right) \boldsymbol{A} \boldsymbol{H} \boldsymbol{W}_V \tag{42}$$

Therefore,

$$\|\mathcal{HC}\left[\mathrm{SA}(\boldsymbol{X})\right]\|_F = \left\| \left(\boldsymbol{I} - \frac{1}{n} \boldsymbol{11}^T\right) \boldsymbol{A} \boldsymbol{H} \boldsymbol{W}_V \right\|_F \tag{43}$$

$$\leq \left\| \boldsymbol{I} - \frac{1}{n} \boldsymbol{11}^T \right\|_2 \|\mathrm{softmax}(\boldsymbol{P})\|_2 \|\boldsymbol{W}_V\|_2 \|\boldsymbol{H}\|_F \tag{44}$$

$$\leq \sqrt{\|\mathrm{softmax}(\boldsymbol{P})\|_1 \|\mathrm{softmax}(\boldsymbol{P})\|_\infty} \|\boldsymbol{W}_V\|_2 \|\boldsymbol{H}\|_F \tag{45}$$

$$= \sqrt{\|\mathrm{softmax}(\boldsymbol{P})\|_1} \|\boldsymbol{W}_V\|_2 \|\boldsymbol{H}\|_F \tag{46}$$

The Eqn. 45 leverages a special case of Hölder's inequality, and the Eqn. 46 can be yielded from $\|\mathrm{softmax}(\boldsymbol{P})\|_\infty = 1$. Now we need to upper bound $\|\mathrm{softmax}(\boldsymbol{P})\|_1$. Suppose $\alpha = \max_{i,j} |\boldsymbol{P}_{ij}|$, then for each $i = 1, \cdots, n$, we have the following inequality for the element with the largest value (say the $j$-th column):

$$\boldsymbol{A}_{ij} = \frac{e^{\boldsymbol{P}_{ij}}}{\sum_{t=1}^n e^{\boldsymbol{P}_{it}}} \leq \frac{e^\alpha}{e^\alpha + \sum_{t \neq j} e^{-\alpha}} = \frac{e^{2\alpha}}{e^{2\alpha} + (n-1)} \tag{47}$$

Hence, we have $\|\mathrm{softmax}(\boldsymbol{P})\|_1 \leq \sum_i \max_j \boldsymbol{A}_{ij} \leq \frac{ne^{2\alpha}}{e^{2\alpha} + (n-1)}$. Insert this result to Eqn. 46, we can conclude the proof. In particular, when $\boldsymbol{P} = \boldsymbol{X} \boldsymbol{W}_Q (\boldsymbol{X} \boldsymbol{W}_K)^T / \sqrt{d}$,

$$\alpha = \max_{i,j} |\boldsymbol{P}_{ij}| = \max_{i,j} \left| \frac{\boldsymbol{x}_i^T \boldsymbol{W}_Q \boldsymbol{W}_K^T \boldsymbol{x}_j}{\sqrt{d}} \right|. \tag{48}$$

Since $\|\boldsymbol{x}_i\|_2, \|\boldsymbol{x}_j\|_2 \leq \gamma, \forall i, j, \alpha \leq \max_{i,j} \|\boldsymbol{x}_i\|_2 \|\boldsymbol{W}_Q \boldsymbol{W}_K^T\|_2 \|\boldsymbol{x}_j\|_2 / \sqrt{d} \leq \gamma^2 \|\boldsymbol{W}_Q \boldsymbol{W}_K^T\|_2 / \sqrt{d}$. □

## C EXTENSION OF THEOREM 3

### C.1 MULTI-HEAD ATTENTION

**Proposition 4.** *(smoothening rate with MSA) Let $\boldsymbol{A}^h = \mathrm{softmax}(\boldsymbol{P}^h)$, where $\boldsymbol{P}^h \in \mathbb{R}^{n \times n}$ with $h = 1, \cdots, H$. Let $\alpha = \max_{h=1}^H \max_{i,j} |\boldsymbol{P}_{ij}^h|$. Define $\mathrm{MSA}(\boldsymbol{X}) = \sum_{h=1}^H \boldsymbol{A}^h \boldsymbol{X} \boldsymbol{W}_V^h \boldsymbol{W}_O^h$ as the output of a multi-head self-attention module, then*

$$\|\mathcal{HC}\left[\mathrm{MSA}(\boldsymbol{X})\right]\|_F \leq \sigma_1 \sigma_2 H \sqrt{\frac{ne^{2\alpha}}{e^{2\alpha} + n - 1}} \|\mathcal{HC}\left[\boldsymbol{X}\right]\|_F,$$

*where $H$ is the number of heads, $\sigma_1 = \max_{h=1}^H \|\boldsymbol{W}_V^h\|_2$ and $\sigma_2 = \max_{h=1}^H \|\boldsymbol{W}_O^h\|_2$.*

*Proof.* For the $h$-th head, according to Theorem 3:

$$\|\mathcal{HC}\left[\text{SA}_h(\boldsymbol{X})\right]\|_F \leq \sqrt{\frac{ne^{2\alpha_h}}{e^{2\alpha_h}+n-1}}\|\boldsymbol{W}_V^h\|_2\|\mathcal{HC}\left[\boldsymbol{X}\right]\|_F, \tag{49}$$

where $\alpha_h = \max_{i,j}|\boldsymbol{P}_{ij}^h|$. Then we have:

$$\|\mathcal{HC}\left[\text{MSA}_h(\boldsymbol{X})\right]\|_F = \left\|\mathcal{HC}\left[\sum_{h=1}^H \text{SA}_h(\boldsymbol{X})\boldsymbol{W}_O^h\right]\right\|_F \tag{50}$$

$$\leq \sum_{h=1}^H \left\|\mathcal{HC}\left[\text{SA}_h(\boldsymbol{X})\boldsymbol{W}_O^h\right]\right\|_F \tag{51}$$

$$\leq \sum_{h=1}^H \sqrt{\frac{ne^{2\alpha_h}}{e^{2\alpha_h}+n-1}}\|\boldsymbol{W}_V^h\|_2\|\mathcal{HC}\left[\boldsymbol{X}\right]\|_2 \tag{52}$$

$$\leq \sum_{h=1}^H \sqrt{\frac{ne^{2\alpha_h}}{e^{2\alpha_h}+n-1}}\|\boldsymbol{W}_V^h\|_2\|\boldsymbol{W}_O^h\|_2\|\mathcal{HC}\left[\boldsymbol{X}\right]\|_F \tag{53}$$

$$\leq \sigma_1\sigma_2 H\sqrt{\frac{ne^{2\alpha}}{e^{2\alpha}+n-1}}\|\mathcal{HC}\left[\boldsymbol{X}\right]\|_F, \tag{54}$$

where Eqn. 51 follows from the linearity of $\mathcal{HC}\left[\cdot\right]$ and triangle inequality. Eqn. 54 can be obtained by relaxing $\alpha_h$, $\|\boldsymbol{W}_V^h\|_2$, and $\|\boldsymbol{W}_O^h\|_2$ to $\alpha$, $\sigma_1$ and $\sigma_2$ (Note that $\sqrt{\frac{ne^{2\alpha}}{e^{2\alpha}+n-1}}$ is monotonically increasing with $\alpha$). $\qquad\square$

## C.2 RESIDUAL CONNECTION

**Proposition 5.** *(smoothening rate with skip connection) Let* $\boldsymbol{A}^h = \text{softmax}(\boldsymbol{P}^h)$, *where* $\boldsymbol{P}^h \in \mathbb{R}^{n\times n}$ *with* $h = 1,\cdots,H$. *Let* $\alpha = \max_{h=1}^H \max_{i,j}|\boldsymbol{P}_{ij}^h|$. *Define* $\boldsymbol{X}' = \text{MSA}(\boldsymbol{X}) + \boldsymbol{X}$ *as the output of a multi-head self-attention module with skip connection, then*

$$\|\mathcal{HC}\left[\boldsymbol{X}'\right]\|_F \leq \left(1 + \sigma_1\sigma_2 H\sqrt{\frac{ne^{2\alpha}}{e^{2\alpha}+n-1}}\right)\|\mathcal{HC}\left[\boldsymbol{X}\right]\|_F$$

*where* $H$ *is the number of heads,* $\sigma_1 = \max_{h=1}^H \|\boldsymbol{W}_V^h\|_2$ *and* $\sigma_2 = \max_{h=1}^H \|\boldsymbol{W}_O^h\|_2$.

*Proof.* By Proposition 4,

$$\|\mathcal{HC}\left[\boldsymbol{X}'\right]\|_F = \|\mathcal{HC}\left[\text{MSA}(\boldsymbol{X}) + \boldsymbol{X}\right]\|_F \tag{55}$$

$$\leq \|\mathcal{HC}\left[\text{MSA}(\boldsymbol{X})\right]\|_F + \|\mathcal{HC}\left[\boldsymbol{X}\right]\|_F \tag{56}$$

$$\leq \sqrt{\frac{ne^{2\alpha}}{e^{2\alpha}+n-1}}\|\mathcal{HC}\left[\boldsymbol{X}\right]\|_F + \|\mathcal{HC}\left[\boldsymbol{X}\right]\|_F \tag{57}$$

$$= \left(1 + \sigma_1\sigma_2 H\sqrt{\frac{ne^{2\alpha}}{e^{2\alpha}+n-1}}\right)\|\mathcal{HC}\left[\boldsymbol{X}\right]\|_F. \tag{58}$$

Again, Eqn. 56 follows from the linearity of $\mathcal{HC}\left[\cdot\right]$ and triangle inequality. $\qquad\square$

## C.3 FEED-FORWARD NETWORK

**Proposition 6.** *(smoothening rate with FFN) Let* $\boldsymbol{A}^h = \text{softmax}(\boldsymbol{P}^h)$, *where* $\boldsymbol{P}^h \in \mathbb{R}^{n\times n}$ *with* $h = 1,\cdots,H$. *Let* $\alpha = \max_{h=1}^H \max_{i,j}|\boldsymbol{P}_{ij}^h|$. *Define* $\boldsymbol{Y} = \text{FFN}(\text{MSA}(\boldsymbol{X}) + \boldsymbol{X})$ *as the output of a transformer block, then*

$$\|\mathcal{HC}\left[\boldsymbol{Y}\right]\|_F \leq \sigma_3\left(1 + \sigma_1\sigma_2 H\sqrt{\frac{ne^{2\alpha}}{e^{2\alpha}+n-1}}\right)\|\mathcal{HC}\left[\boldsymbol{X}\right]\|_F$$

*where* $\text{FFN} : \mathbb{R}^d \to \mathbb{R}^d$ *represents a feed-forward network,* $H$ *is the number of heads,* $\sigma_1 = \max_{h=1}^H \|\boldsymbol{W}_V^h\|_2$, $\sigma_2 = \max_{h=1}^H \|\boldsymbol{W}_O^h\|_2$, *and* $\sigma_3 = \text{Lips}(\text{FFN})$ *is the Lipschitz constant of FFN. In particular,* $\sigma_3 = 1 + \text{Lips}(\text{FFN})$ *when residual connection is considered in FFN.*

*Proof.* Let $\boldsymbol{X}' = \mathrm{MSA}(\boldsymbol{X}) + \boldsymbol{X}$ and $\boldsymbol{z}' = \mathbf{1}^T(\boldsymbol{X}'/n) \in \mathbb{R}^d$, then we have

$$\|\mathcal{HC}\left[\mathrm{FFN}(\boldsymbol{X}')\right]\|_F \leq \|\mathrm{FFN}(\boldsymbol{X}') - \mathbf{1}\,\mathrm{FFN}(\boldsymbol{z})^T\|_F \tag{59}$$

$$= \|\mathrm{FFN}(\boldsymbol{X}') - \mathrm{FFN}(\mathbf{1}\boldsymbol{z}^T)\|_F \tag{60}$$

$$\leq \sigma_3 \|\boldsymbol{X}' - \mathbf{1}\boldsymbol{z}^T\|_F \tag{61}$$

$$= \sigma_3 \left\|\left(\boldsymbol{I} - \frac{1}{n}\mathbf{1}\mathbf{1}^T\right)\boldsymbol{X}'\right\|_F = \sigma_3 \|\mathcal{HC}\left[\boldsymbol{X}'\right]\|_F \tag{62}$$

$$\leq \sigma_3 \left(1 + \sigma_1\sigma_2 H\sqrt{\frac{ne^{2\alpha}}{e^{2\alpha}+n-1}}\right)\|\mathcal{HC}\left[\boldsymbol{X}\right]\|_F, \tag{63}$$

where Eqn. 59 follows from Lemma 7, Eqn. 60 holds because FFN operates row-wisely on feature matrix, and Eqn. 61 is due to the definition of Lipschitz constant. Finally, Eqn. 63 is yielded from Proposition 5. $\qquad\square$

**Lemma 7.** *Given $\boldsymbol{X} \in \mathbb{R}^{n \times d}$, $\|\mathcal{HC}\left[\boldsymbol{X}\right]\|_F \leq \|\boldsymbol{X} - \mathbf{1}\boldsymbol{z}^T\|_F$ for all $\boldsymbol{z} \in \mathbb{R}^d$.*

*Proof.* We prove the Lemma by showing that $\boldsymbol{z}^* = \boldsymbol{X}^T\mathbf{1}/n$ achieves the minimum of the optimization problem $\arg\min_{\boldsymbol{z}}\|\boldsymbol{X} - \mathbf{1}\boldsymbol{z}^T\|_F^2$.

$$\|\boldsymbol{X} - \mathbf{1}\boldsymbol{z}^T\|_F^2 = \mathrm{Tr}(\boldsymbol{X}^T - \boldsymbol{z}\mathbf{1}^T)(\boldsymbol{X} - \mathbf{1}\boldsymbol{z}^T) \tag{64}$$

$$= \mathrm{Tr}(\boldsymbol{X}^T\boldsymbol{X}) - \mathrm{Tr}(\boldsymbol{z}\mathbf{1}^T\boldsymbol{X}) - \mathrm{Tr}(\boldsymbol{X}^T\mathbf{1}\boldsymbol{z}^T) + \mathrm{Tr}(\boldsymbol{z}^T\mathbf{1}^T\mathbf{1}\boldsymbol{z}) \tag{65}$$

$$= n\boldsymbol{z}^T\boldsymbol{z} - 2\,\mathrm{Tr}(\boldsymbol{X}^T\mathbf{1}\boldsymbol{z}^T) + \mathrm{Tr}(\boldsymbol{X}^T\boldsymbol{X}) \tag{66}$$

It is easy to show the derivative in terms of $\boldsymbol{z}$:

$$\nabla_{\boldsymbol{z}}\|\boldsymbol{X} - \mathbf{1}\boldsymbol{z}^T\|_F^2 = 2n\boldsymbol{z} - 2\boldsymbol{X}^T\mathbf{1}. \tag{67}$$

Therefore, $\boldsymbol{z}^* = \frac{1}{n}\boldsymbol{X}^T\mathbf{1}$ achieves the minimum. $\qquad\square$

## D  DEFERRED REMARKS ON SECTION 2.4

### D.1  CONNECTION WITH RANDOM WALK THEORY

We add that the asymptotic evolution of feature representations can be interpreted through the lens of random walk theory. We can regard self-attention map $\boldsymbol{A}$ as probability transition matrices for a Markov chain. Since each entry is larger than zero, the Markov chain should be irreducible. This implies the Markov chain will converge into a unique stationary distribution $\boldsymbol{\pi}$ (Randall, 2006). Let $\boldsymbol{a}_i$ denote the $i$-th row of $\boldsymbol{A}$, then for all $i = 1, \cdots, n$ we have $\|(\boldsymbol{a}_i)^T\boldsymbol{A} - \boldsymbol{\pi}^T\| \leq \lambda\|\boldsymbol{a}_i - \boldsymbol{\pi}\|$, where $\lambda \in (0,1)$ is the mixing rate of the transition matrix $\boldsymbol{A}$. As a consequence, $\lim_{l\to\infty}\boldsymbol{A}^l = \mathbf{1}\boldsymbol{\pi}^T$ yields a pure low-pass filter. When repeatedly applying this self-attention matrix to feature maps, $\lim_{l\to\infty}\boldsymbol{A}^l\boldsymbol{X} \to \mathbf{1}\boldsymbol{\pi}^T\boldsymbol{X}$ only preserves the rank-1/DC portion of the signals, which is consistent with our Theorem 1. Nevertheless, this interpretation does not bring other transformer components into consideration. And our theory further provides a concrete convergence rate with respect to the network parameters (Theorem 3).

### D.2  REMARKS ON DONG ET AL. (2021)

Here we respectfully elaborate on the hidden assumptions in the proof of the current preprint of Dong et al. (2021).

1) In the proof of Lemma A.3, Taylor expansion was used to approximate and upper bound an exponentiation. However, to let the right-hand side upper bound satisfied, we conjectured that the authors implicitly assumed $\boldsymbol{E}_{ij} - \boldsymbol{E}_{ij'}$ is bounded around zero. After directly communicating with the authors, they confirmed that a missed assumption here is $\max_{i,j}(\boldsymbol{E}_{ij} - \boldsymbol{E}_{ij'}) \leq 1$.

2) In the proof of Lemma A.1, to let Eqn. (8)-(9) hold, the authors may have assumed $\boldsymbol{R}, \boldsymbol{W}_V \geq 0$, where $\geq$ denotes entry-wise inequality. As the authors suggested, an entry-wise absolute value can be imposed to $\boldsymbol{R}$ and $\boldsymbol{W}_V$ as a simple fix, without influencing their $\ell_1$ and $\ell_\infty$ norm. However,

even after those changes, we still have difficulty walking through Eqn. (6)-(8), and we are currently communicating with the authors on this matter.

3) In the proof of Lemma A.1, we find Eqn. (12) may not be satisfied in general. We can raise the following counterexample: Since $\boldsymbol{E}, \boldsymbol{r}, \boldsymbol{R}, \boldsymbol{W}_V$ can be any matrices, we simply let $\boldsymbol{D} = \text{diag}(2,3), \text{softmax}(\boldsymbol{r}) = \begin{bmatrix} 0.8 & 0.2 \end{bmatrix}^T$, $\boldsymbol{R} = \boldsymbol{W}_V = \boldsymbol{I}$. Then $\|\boldsymbol{D}\mathbf{1}\text{softmax}(\boldsymbol{r})^T \boldsymbol{R} \boldsymbol{W}_V\|_1 = 4$ while $\|\boldsymbol{D}\mathbf{1}\|_\infty \|\boldsymbol{R}\|_1 \|\boldsymbol{W}_V\|_1 = 3$, which disproves the claim. We conjecture that some additional prerequisite constraints on $\boldsymbol{E}, \boldsymbol{r}$ might be needed here to proceed the derivation, and we are currently communicating with the authors on this matter.

### D.3 Generalize to Other Attention Mechanism

Our theorizing can be smoothly generalized to other attention mechanisms because our Theorem 1 and 3 do not require any prior knowledge on pre-softmax pairwise correlation $\boldsymbol{P}$.

**Logistic Attention.** We refer logistic attention to the attention mechanism used in Veličković et al. (2018); Verma et al. (2018), where attention is calculated via a linear combination:

$$\boldsymbol{A}_{ij} = \frac{\exp\left(\boldsymbol{x}_i^T \boldsymbol{u}_Q + \boldsymbol{x}_j^T \boldsymbol{u}_K + b\right)}{\sum_t \exp\left(\boldsymbol{x}_i \boldsymbol{u}_Q + \boldsymbol{x}_t \boldsymbol{u}_K + b\right)} \tag{68}$$

where $\boldsymbol{u}_Q$ and $\boldsymbol{u}_K$ are query/key parameters, $b$ is the bias term. With the same condition in Theorem 3, we can upper bound $\alpha$ by $|(\|\boldsymbol{u}_K\|_2 + \|\boldsymbol{u}_Q\|_2)\gamma + b|$.

**L2 Distance Attention.** L2 distance based attention (Kim et al., 2021) Lipschitz formulation of self-attention. The pair-wise attention can be written as follows:

$$\boldsymbol{A}_{ij} = \frac{\exp\left(-\|\boldsymbol{x}_i^T \boldsymbol{W}_Q - \boldsymbol{x}_j^T \boldsymbol{W}_K\|_2^2/\tau\right)}{\sum_t \exp\left(-\|\boldsymbol{x}_i^T \boldsymbol{W}_Q - \boldsymbol{x}_t^T \boldsymbol{W}_K\|_2^2/\tau\right)} \tag{69}$$

where $\boldsymbol{W}_Q$ and $\boldsymbol{W}_k$ are query/key weights, and $\tau$ is a scaling factor. Similar to Theorem 3, we can upper bound $\alpha$ by $(\|\boldsymbol{W}_K\|_2 + \|\boldsymbol{W}_Q\|_2)^2 \gamma^2/\tau$.

## E An Auxiliary Lemma for AttnScale

**Lemma 8.** *Let $\tilde{\boldsymbol{A}} = \mathcal{F} \boldsymbol{A} \mathcal{F}^{-1}$ be the spectral response of attention matrix $\boldsymbol{A}$, and parameterize a low-filter by $\boldsymbol{L} = \mathcal{F}^{-1} \text{diag}(\beta, 0, \cdots, 0)\mathcal{F}$. Then $\beta^* = 1$ is the optimal solution of the following optimization problem:* $\arg\min_\beta \|\boldsymbol{A} - \boldsymbol{L}\|_F$.

*Proof.* First we make simplification $\boldsymbol{L} = \mathcal{F}^{-1} \text{diag}(\beta, 0, \cdots, 0)\mathcal{F} = \beta \mathbf{1}\mathbf{1}^T$ (refer to Appendix A). Then we have:

$$\|\boldsymbol{A} - \boldsymbol{L}\|_F^2 = \|\boldsymbol{A} - \beta\mathbf{1}\mathbf{1}^T\|_F^2 = \text{Tr}(\boldsymbol{A} - \beta\mathbf{1}\mathbf{1}^T)^T(\boldsymbol{A} - \beta\mathbf{1}\mathbf{1}^T) \tag{70}$$

$$= \text{Tr}\left(\boldsymbol{A}^T\boldsymbol{A} - \frac{\beta}{n}\boldsymbol{A}^T\mathbf{1}\mathbf{1}^T - \frac{\beta}{n}\mathbf{1}\mathbf{1}^T\boldsymbol{A} + \frac{\beta^2}{n}\mathbf{1}\mathbf{1}^T\right) \tag{71}$$

$$= \frac{\beta^2}{n}\text{Tr}\,\mathbf{1}\mathbf{1}^T - \frac{2\beta}{n}\text{Tr}\,\mathbf{1}\mathbf{1}^T\boldsymbol{A} + \text{Tr}\,\boldsymbol{A}^T\boldsymbol{A} \tag{72}$$

$$= \beta^2 - \frac{2\beta}{n}\sum_{j=1}^n\sum_{i=1}^n \boldsymbol{A}_{ij} + \text{Tr}\,\boldsymbol{A}^T\boldsymbol{A} \tag{73}$$

$$= \beta^2 - 2\beta + \text{Tr}\,\boldsymbol{A}^T\boldsymbol{A} \tag{74}$$

From Eqn. 74, $\beta^* = 1$ achieves the minimum of the objective function. $\square$

## F More on Visualization

### F.1 Details on Figure 1

To verify our Theorem 3, we depict the high-frequency intensity of each layer's output and its theoretical upper bound. Our visualization is based on the official checkpoint of 12-layer DeiT-S. Since

training a ViT without either FFN or residual connection will certainly cause failure, we remove these components directly from the pre-trained model to illustrate the effects of different components. We use logarithmic scale for the purpose of better view. Let $X_l$ denote the output of the $l$-th layer, and $X_0$ be the initial inputs. For red line, we directly calculate $\log(\|\mathcal{HC}\,[X_l]\|_F/\|X_0\|_F)$ at each layer. For blue line, we first obtain the coefficient $\gamma_l$ in Section 2.2 and 2.3 with respect to network parameters (e.g., we can compute $\gamma_l = \sqrt{\frac{ne^{2\alpha}}{e^{2\alpha}+n-1}}\|W_V\|_2$ for attention only architecture). Then we estimate the upper bound by $\gamma_l\|\mathcal{HC}\,[X_{l-1}]\|_F$ and apply the logarithm by $\log(\gamma_l\|\mathcal{HC}\,[X_{l-1}]\|_F/\|X_0\|_F)$. To summarize, one can see without residual connection, the first two sub-figures imply an exponential convergence rate, which is consistent with our Theorem 3.

## F.2 MORE VISUALIZATION ON SPECTRUM

In this appendix, we provide more visualization on the spectrum of attention map to validate our Theorem 1. We compute the spectrum of attention map $A$ for both Fig. 2 and Fig. 5 in the following way. By regarding $A$ as a linear filter, its Fourier-domain response is another linear kernel $\Lambda = \mathcal{F}A\mathcal{F}^{-1}$. When $\Lambda$ is applied to a spectrum $\tilde{x} = \mathcal{F}x$ of signals $x$, the $i$-th frequency response will be $\Lambda_i\tilde{x}$, where $\Lambda_i$ is the $i$-th row of $\Lambda$. Hence, we can use $\|\Lambda_i\|_2$ to evaluate the spectral response intensity of the $i$-th frequency band. Below we provide a complete spectral visualization of attention maps computed from a random sample in ImageNet validation set.

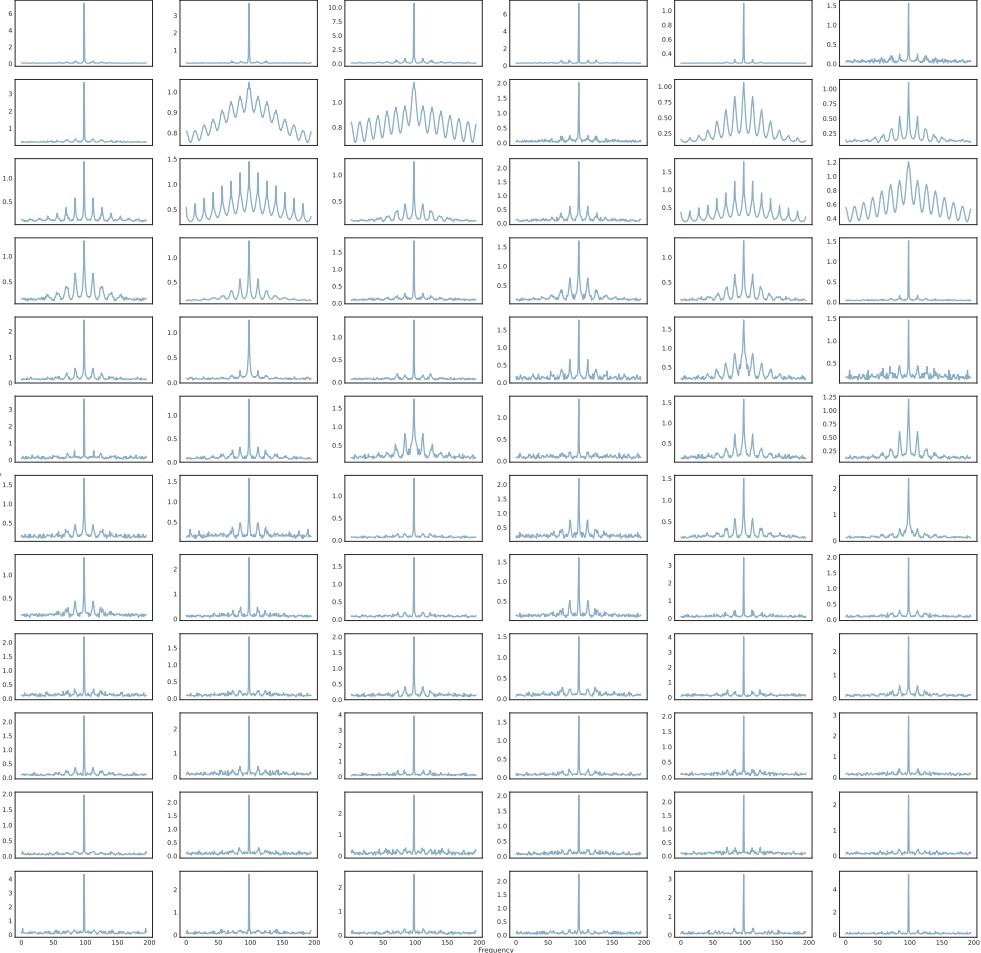

Figure 5: Visualize the spectrum of attention maps. Each row demonstrates every head at a same layer, and from top to bottom, the 12 rows correspond to 1 ~ 12-th layer, for left to right, the 6 columns correspond to 1 ~ 6-th head, respectively. Best view in a zoomable electronic copy.

### F.3 DETAILS ON SIMILARITY CURVES (FIGURE 4)

In Fig. 4, we visualize the cosine similarity of attention maps and feature maps to show the effectiveness of our AttnScale and FeatScale on 24-layer DeiT, respectively. We follow the definition in Zhou et al. (2021a) to compute the cosine similarity metric for attention maps. Instead of measuring cross-layer similarity, we calculate average cross-patch similarity at the same layer. Given the layer index $l$ and corresponding attention maps $\boldsymbol{A}^{(l,h)} \in \mathbb{R}^{n \times n}$, the cosine similarity can be computed by:

$$M_{\text{attn}}^l = \frac{2}{n(n-1)H} \sum_{h=1}^{H} \sum_{i=1}^{n} \sum_{j=i+1}^{n} \frac{\left| \boldsymbol{A}_{:,i}^{(l,h)T} \boldsymbol{A}_{:,j}^{(l,h)} \right|}{\left\| \boldsymbol{A}_{:,i}^{(l,h)} \right\|_2 \left\| \boldsymbol{A}_{:,j}^{(l,h)} \right\|_2}, \tag{75}$$

where $\boldsymbol{A}_{:,i}^{(l,h)}$ denotes the $i$-th column of $\boldsymbol{A}^{(l,h)}$, and $H$ is the number of heads. The cosine similarity between $i$-th and $j$-th column of $\boldsymbol{A}^{(l,h)}$ measures how the contribution of one token (say the $i$-th token) varies from the other (say the $j$-th token). We average the similarity between every pair of tokens' attention map (excluding the self-to-self similarity) and every attention head. We refer interested readers to Zhou et al. (2021a) for more details.

We use the similar metric to compute similarity for feature maps. Following Gong et al. (2021), we compute pair-wise cosine similarity between every two different tokens. Formally, given the layer index $l$, and its output $\boldsymbol{X}^{(l)} \in \mathbb{R}^{n \times d}$, the cosine similarity is estimated by:

$$M_{\text{feat}}^l = \frac{2}{n(n-1)} \sum_{i=1}^{n} \sum_{j=i+1}^{n} \frac{\left| \boldsymbol{X}_{i,:}^{(l)T} \boldsymbol{X}_{j,:}^{(l)} \right|}{\left\| \boldsymbol{X}_{i,:}^{(l)} \right\|_2 \left\| \boldsymbol{X}_{j,:}^{(l)} \right\|_2}, \tag{76}$$

where $\boldsymbol{X}_{i,:}^{(l)}$ denotes the $i$-th row of $\boldsymbol{X}^{(l)}$. The cosine similarity between between $i$-th and $j$-th row of $\boldsymbol{X}^{(l)}$ measures how similar the feature representations of two tokens are. Likewise, we average the similarity between every pair of tokens' features except for the self-to-self similarity. More details can found in Gong et al. (2021). We additionally provide a visualization of these two metrics for 12-layer DeiT in Fig. 12.

## G DEFERRED EXPERIMENTS AND MODEL INTERPRETATION

### G.1 FINE-TUNING EXPERIMENTS

Our deferred fine-tuning experiment with CaiT (Touvron et al., 2021b) results are presented in Table 3. Different from trining scratch, we fine-tune CaiT with AttnScale and FeatScale parameters from the pre-trained models for 60 epochs following Gong et al. (2021). For a fair comparison, we simultaneously train a plain CaiT for another 60 epochs. During fine-tuning, we reduce learning rate to $5 \times 10^{-5}$ and weight decay to $5 \times 10^{-4}$. All other hyper-parameters and training recipe are kept consistent with the original paper (Touvron et al., 2021b).

Table 3: Experimental evaluation of finetuning AttnScale & FeatScale with CaiT. The number inside the ($\uparrow \cdot$) represents the performance gain compared with the baseline model, and accuracies within/out of parenthesis are the reported/reproduced performance.

| Backbone | Method | Input size | # Layer | # Param | FLOPs | Throughput | Top-1 Acc (%) |
|---|---|---|---|---|---|---|---|
| CaiT | CaiT-XXS | 224 | 24 | 12.0M | 2.53G | 589.3 | 77.5 (77.6) |
| | CaiT-XXS + AttnScale | 224 | 24 | 12.0M | 2.53G | 548.1 | 77.8 ($\uparrow$ 0.3) |
| | CaiT-XXS + FeatScale | 224 | 24 | 12.0M | 2.53G | 573.5 | 77.8 ($\uparrow$ 0.3) |
| | CaiT-S | 224 | 24 | 46.9M | 8.74G | 371.9 | 82.6 (82.7) |
| | CaiT-S + AttnScale | 224 | 24 | 46.9M | 8.75G | 339.0 | 82.8 ($\uparrow$ 0.2) |
| | CaiT-S + FeatScale | 224 | 24 | 46.9M | 8.75G | 358.2 | 82.9 ($\uparrow$ 0.3) |

### G.2 VISUALIZATION AND INTERPRETATION OF ATTNSCALE

In this subsection, we provide visualization to interpret our AttnScale and further support our experiments. ❶ In Fig. 6 we visualize the learned weights of our AttnScale. We observe conclude our AttnScale are successfully trained to amplify the high-pass component. We also find when layer

index goes larger, the scaling weights turns larger to prevent attention collapse at deeper layer. ❷ We also compare the attention map produced by AttnScale with those produced by original DeiT. We observe from Fig. 8 that our AttnScale can extract more salient and higher contrastive attention than vanilla DeiT, which indicates our AttnScale possesses higher capability to distinguish tokens from larger variety of attention schemes. ❸ To be more objective, we plot the spectrum of a 24-layer DeiT's attention maps with/without our AttnScale in Fig. 9 and 10. The visualization procedure has been elaborated in Sec. F.2. We find that attention maps from AttnScale enjoy richer filtering diversity, capable of performing high-pass (row 2, column 3) and band-pass (row 12, column 5) filtering, instead of only low-pass filtering (see Fig. 9).

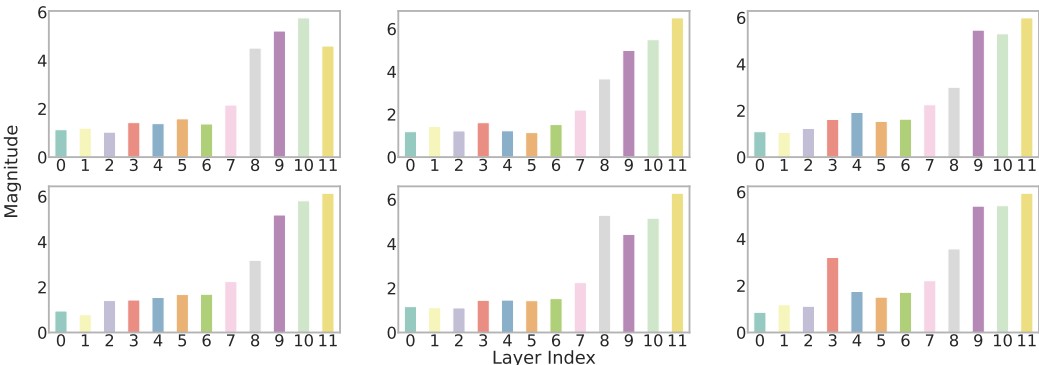

Figure 6: Visualize the learned weights of DeiT-S + AttnScale. Each sub-plot depicts the scaling weights of the same head for different layers. For left to right, top to bottom, six sub-figures correspond to 1 ~ 6-th head, respectively. Best view in color.

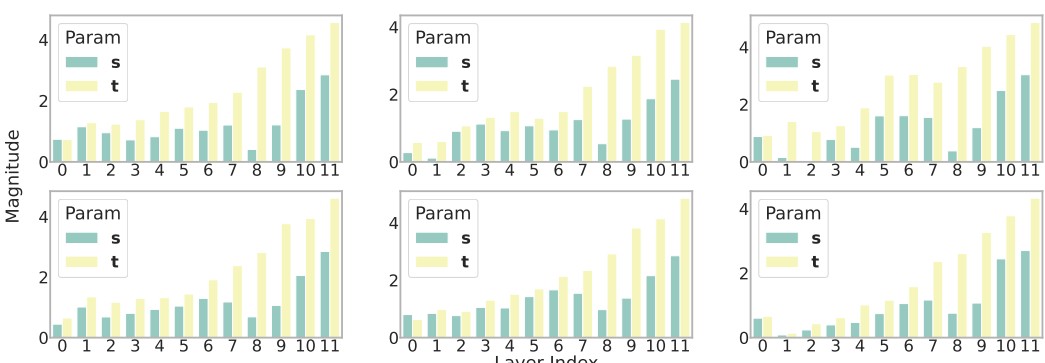

Figure 7: Visualize the learned weights of DeiT-S + FeatScale. Each sub-plot depicts two groups of scaling weights of the same head for different layers. For left to right, top to bottom, six sub-figures correspond to 1 ~ 6-th head, respectively. Best view in color.

### G.3 VISUALIZATION AND INTERPRETATION OF FEATSCALE

In this subsection, we provide visualization to interpret our FeatScale. ❶ We plot the scaling weights of FeatScale in Fig. 7. We observe that the re-weighting factors learned for high-frequency components $t$ is consistently larger than the weights for the DC term $s$, which indicates our FeatScale is successfully trained to elevate high-frequency features against the dominance of DC component. Similarly, the gap between $s$ and $t$ becomes huger when going deeper. ❷ We also demonstrate the proportion of feature maps' high-frequency component in Fig. 11 for both 12 (lower one) / 24(upper one) -layer DeiT. The proportion value is calculated by $\|\mathcal{HC}[\boldsymbol{X}]\|_F / \|\boldsymbol{X}\|_F$. We find high-frequency signals diminish quickly at deeper layer, and 24-layer DeiT suffers from a faster pace. Our FeatScale is effective to keep the high-frequency signals stand for both 12-layer and 24-layer DeiT.

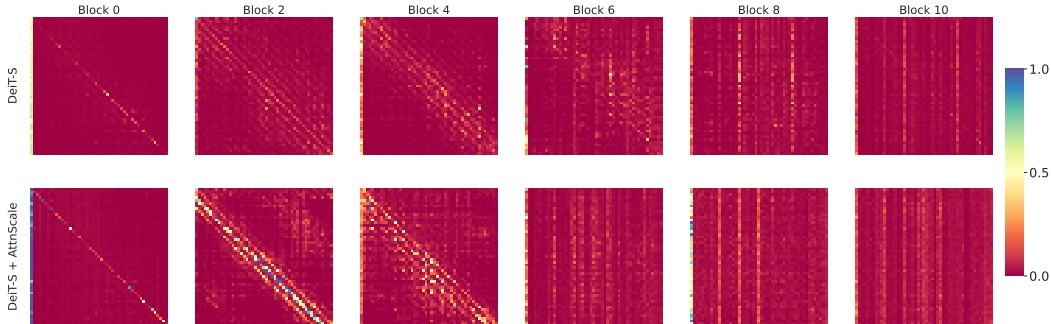

Figure 8: Visualize the attention map of DeiT-S with/without AttnScale. $4 \times 4$ max pooling has been applied. The first row visualizes attention maps without AttnScale, and the second row visualizes attention maps with AttnScale. Each column corresponds to the layer noted by its sub-title. The attention map are computed from a random sample in ImageNet validation set. We only demonstrate the first head of each layer. Best view in a zoomable electronic copy.

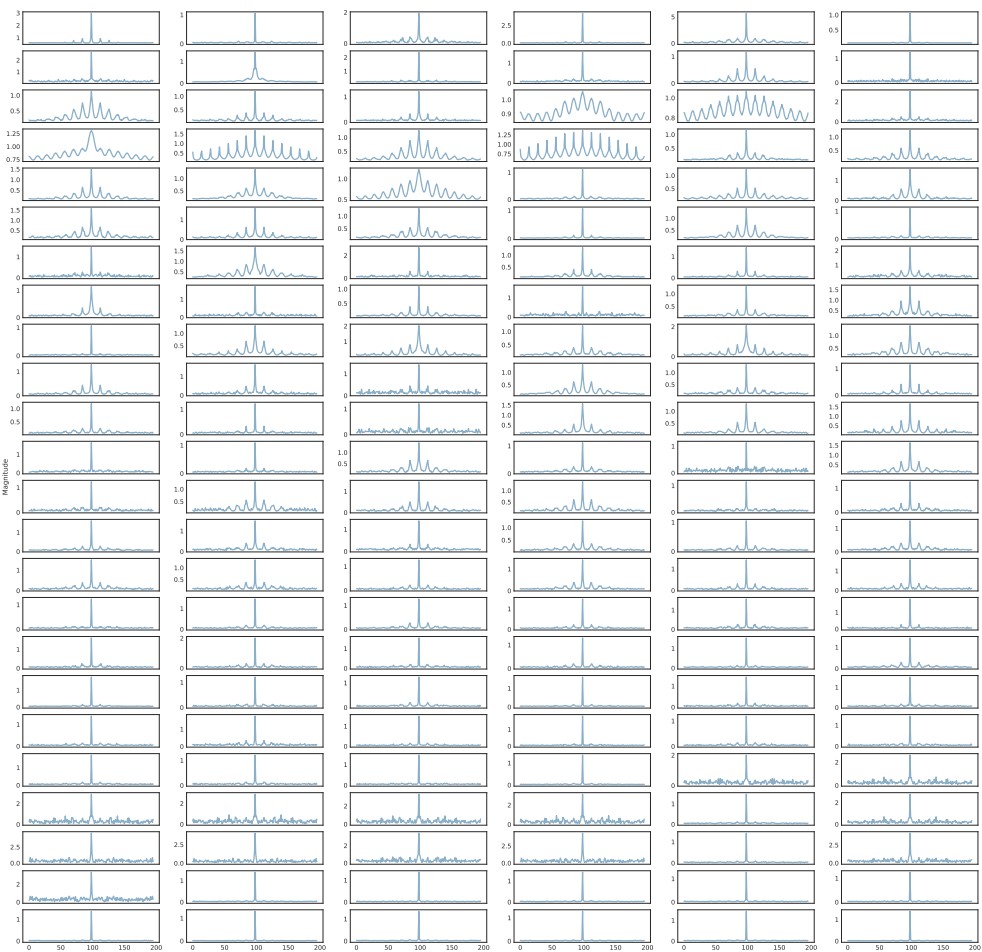

Figure 9: Visualize the spectrum of attention maps **without AttnScale**. Each row demonstrates every head at a same layer, and from top to bottom, the 24 rows correspond to 1 ~ 24-th layer, for left to right, the 6 columns correspond to 1 ~ 6-th head, respectively. Best view in a zoomable electronic copy.

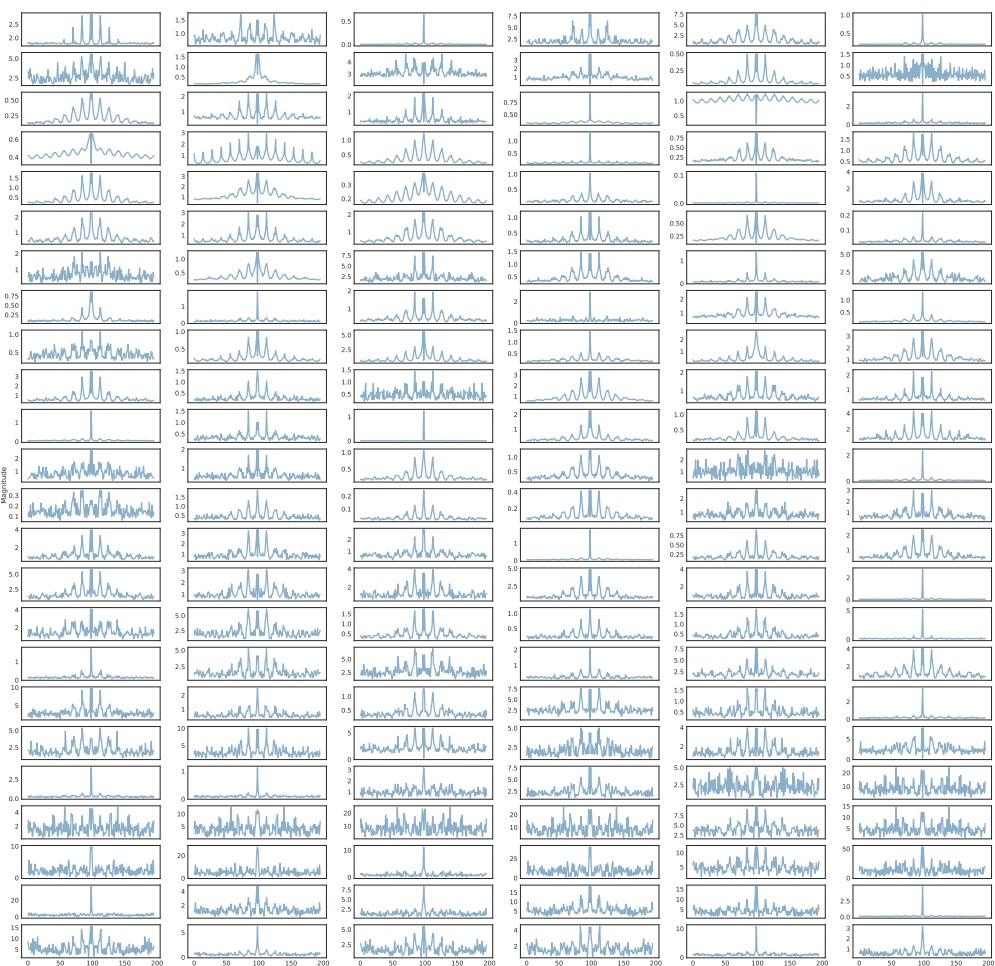

Figure 10: Visualize the spectrum of attention maps **with AttnScale**. Each row demonstrates every head at a same layer, and from top to bottom, the 24 rows correspond to 1 ~ 24-th layer, for left to right, the 6 columns correspond to 1 ~ 6-th head, respectively. Best view in a zoomable electronic copy.

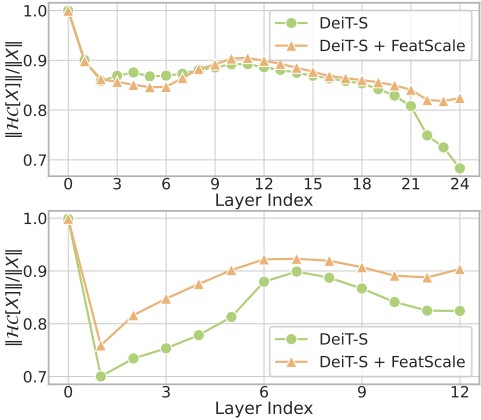

Figure 11: Visualize the proportion of the high-frequency component of feature maps with/without our FeatScale on 12/24 layer DeiT. Refer to Appendix G.3 for details.

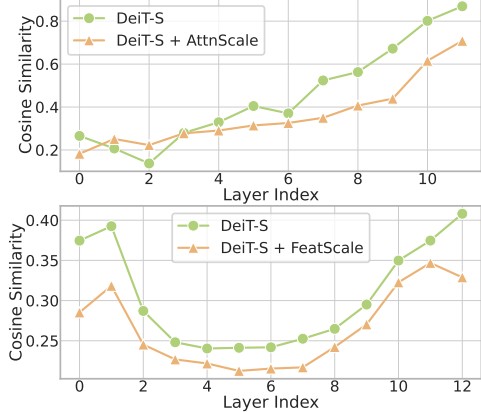

Figure 12: Visualize cosine similarity of attention and feature maps with/without our proposed methods on 12-layer DeiT. Refer to Appendix F.3 for details.

