# OpenReview forum: "Anti-Oversmoothing in Deep Vision Transformers via the Fourier Domain Analysis: From Theory to Practice"
_ICLR.cc/2022/Conference — ICLR 2022 Poster_

### Official Review · Reviewer_kYZM · 2021-11-02

**Correctness:** 3
**Technical Novelty And Significance:** 3
**Empirical Novelty And Significance:** 2
**Recommendation:** 6
**Confidence:** 4

**Main Review:**

Strengths:
1.	The paper provides theoretical analysis of ViT from spectral domain, which explains the empirically findings of patch feature collapse of ViTs.
2.	Reweight techniques of low-pass and high-pass components are proposed, which is sound according to the theory part.

Weaknesses:
Experiments part does not strongly support the proposed method.
1.	The gain is limited, 0.2 ~ 0.3 improvements for 24-layer models.
2.	With FeatScale, 12-layer DeiT has 1.0 improvement v.s. 24-layer DeiT has 0.3 improvement, this is kind of opposite to the paper’s claim, as layers increases, the collapse should be more severe, where one would expect the proposed techniques should be more helpful,  however the experiments results indicates the opposite. It questions the proposed methods’ ability to scaling depths further beyond 24 layers.




------post-rebuttal-----
With the new provided experiments results, I raised my rating towards accept.

**Summary Of The Paper:**

This paper provides a viewpoint of spectrum domain to show self-attention amounts to a low-pass filter and will cause feature maps to only preserve direct-current components as depth increases. Based on this viewpoint, it proposes two technique named AttnScale, FeatScale to reweight low-pass, high-pass components with minimal parameters. Experiments are conducted on ImageNet.

**Summary Of The Review:**


The paper address the feature collapse problem of ViT from spectral prospective, decomposing attention to low-pass and high-pass components. The therotical part is nice, while the experiments does not show adequate evidence that the proposed method would actually help scale depths of ViTs (e.g. beyond 24 layers) as title indicates.

---

> ### Author Response · Authors · 2021-11-14
> **Reply to Reviewer kYZM**
>
> Dear reviewer kYZM:
>
> We appreciate your acknowledgment of the theoretical contributions and practical value of our work. We are excited and confident about the verifiably superior performance in the experiment section, since they are not only strictly better than the state-of-the-art methodologies, but also offer cheap and fast improvements on top of the already well-trained SOTA models.
>
> With regard to your concerns on the experimental results, we respectfully believe that they are out of misinterpretations of our experiments. You mentioned the performance gain for 24 layers is limited, with only 0.2~0.3% improvements, which is not true. As shown in table 1 comparing over 24-layer DeiT, the proposed methods consistently bring 0.6% and 0.8% performance gain instead of 0.3%. The codebase we actually followed is the codebase of DiversePatch [1], which lead to a reproduced result of 80.5% for 24-layer DeiT, instead of 81.0% reported in the CaiT paper. We have clarified this in Sec. 5.1 of the updated PDF. Given the 0.8% improvement over 24 layers, your second concern shouldn’t be an issue: the ability of the proposed method to scale further beyond 24 layers is promising, as the performance boost remains similarly noticeable at 24 layers.
>
> Beyond boosting deeper ViTs, the proposed method has offered a much greater potential: it almost always brings a significant improvement over the trained models with a cheap fine-tune step of 60 epochs, as reported in table 1.
>
> The effectiveness of our model is further validated via two new experiments:
>
> 1. We trained a 24-layer CaiT with AttnScale and FeatScale from scratch. Both of them manage to attain a 0.6% accuracy gain.
>
> 2. We trained a Swin-B with AttnScale and FeatScale. The former one brings 0.4% improvements, and the latter one brings 0.5% improvements.
>
> [1] Gong et al. Vision Transformers with Patch Diversification. https://github.com/ChengyueGongR/PatchVisionTransformer
>
> We sincerely hope our response could lead to a better understanding of the effectiveness of the proposed methodologies and could lead to a fair and positive assessment.
>
> Best,
>
> Authors of paper 887

---

> > ### Author Response · Authors · 2021-11-23
> > **Sincerely expecting further discussions from Reviewer kYZM**
> >
> > Dear Reviewer kYZM:
> >
> > We want to thank you for the constructive comments in your review. As a follow-up on our responses, we would like to kindly remind you that the discussion period is ending soon. We hope to use this open response period to discuss the paper to solve the concerns and improve the quality of our paper. Have you gotten a chance to read our responses and revision, which attempt to address all of your concerns? We have included a couple of new experiments and visualizations to the new manuscript (see Table 1 and Appendix G).
> >
> > We sincerely hope to have further discussions with you to see if our response solves the concerns. We would be more than happy to provide more information or clarification, should it be necessary. We hope our paper, as the first few rigorous analysis on ViT, could be valued and receive a positive and fair assessment.
> >
> > Best,
> >
> > Authors of paper 887

---

> > ### Comment · Reviewer_kYZM · 2021-11-23
> > **Thanks for the response.**
> >
> > Thanks for the response. The explantation for DeiT gain and new provided results of CaiT and Swin models have resolved my concerns. I will raise the rating

---

> > > ### Author Response · Authors · 2021-11-26
> > > **Thank you**
> > >
> > > Dear reviewer kYZM:
> > >
> > > Thank you for the precious endorsement and review!
> > >
> > > Best,
> > >
> > > Authors of paper 887

---

### Official Review · Reviewer_DHWs · 2021-11-04

**Correctness:** 2
**Technical Novelty And Significance:** 3
**Empirical Novelty And Significance:** 2
**Recommendation:** 6
**Confidence:** 5

**Main Review:**

However, some problems remain:
1.The article proves that the self-attention is a low-pass filter, thus neglecting the high-frequency information. The conclusion comes to the stage that pure attention leads to rank collapse in the network. FFN, skip connections and mutli-head self-attention mechanism which are used in ViTs help to hold high-frequency components. The current clarification in the paper are not very convincing to verify why the ViTs cannot go deeper, even though with FFN, skip connections and mutli-head.
2. The visualization of the modified attention is about the learnable parameters. However, the spectrum of the modified attention seems more convincing to show the change on high-frequency components. And it is more persuasive to show the situation when the depth is 24.


**Summary Of The Paper:**

This paper explores the reasons why ViTs cannot go deeper. The article provides clear and solid proofs, clarifying the rank collapse in attention matrix via Fourier analysis. Meanwhile, the authors suggest two types of possible solutions, AttnScale and FeatScale, which are useful when scaling up the ViTs architecture. The experiments on DeiT and CaiT show that the adjustment on attention helps avoiding rank collapse when going deeper, holding high-frequency information.


**Summary Of The Review:**

The overall presentation is good. The main concerns are in above.
The models (CaiT-S and DeiT-S) are fairly small. Can the authors provide more experiments on big models to further verify the effectiveness of the proposed method?

---

> ### Author Response · Authors · 2021-11-14
> **Reply to Reviewer DHWs**
>
> Dear reviewer DHWs:
>
> Thank you for acknowledging the theoretical contribution and the novelty of this paper, and we appreciate your comments to help make our paper stronger. We provide pointwise responses to your concerns about our motivation and experiments below.
>
> ***Q1.*** Why the ViTs cannot go deeper, even though with FFN, skip connections and multi-head?
>
> ***Reply:***
> Thanks for picking this question, which is the key question that this paper targets addressing. We have offered rigorous theoretical analysis in proposition 4/5/6 and offered empirical explanations in the experiment section 5. A short answer to the limited improvement of FFN/skip connections/multi-head towards deeper ViT is: though these mechanisms help preserve the high-frequency signals, they do not change the fact that multi-head self-attention (MSA) block as a whole only possesses the expressive power of low-pass filters. As has been systematically analyzed in theorem 1, MSA is the key culprit that leads to over-smoothness and failure of going deeper.
>
>
> Indeed, our propositions 4, 5, 6 state multi-head, FFN, skip connections can slow down the convergence. However, according to our derivation, this slow-down is brought by simultaneously amplifying the low-frequency and high-frequency signals with the same constant. Multi-head, FFN, skip-connection are incapable of promoting high-frequency information. It is inevitable that high-frequency components are continuously diluted as ViT goes deeper. This restricts the expressiveness of ViT, resulting in the performance saturation in deeper ViT.
>
> We have updated our PDF to include these discussions in section 2.3, please kindly have a check.
>
> ***Q2.*** The spectrum of the modified attention seems more convincing to show the change on high-frequency components.
>
> ***Reply:***
> We appreciate your constructive comments. We have updated the PDF with two visualizations of the modified spectrum for 24-layer ViT in Fig. 8,9 (Appendix G). In summary, with our proposed methods, attention maps indeed can attain more diverse filtering power. We are still actively working on generating more visualization for 24-layer ViT. Upon finishing, we will update them to the PDF and let you know.
>
>
> ***Q3.*** Can the authors provide more experiments on big models to further verify the effectiveness of the proposed method?
>
>
> ***Reply:***
> We appreciate your suggestion to strengthen the empirical perspective of this paper. We have added an experiment where we combined our AttnScale and FeatScale with Swin-S (50M params). The new results are presented in section 5.1 table 1 of the updated PDF. In brief, the result demonstrates that our AttnScale and FeatScale can boost Swin-S by 0.4% and 0.5% in top-1 accuracy, respectively, which achieve the SOTA performance (83.5%), comparable to Swin-B (88M params), with only hundreds of parameter overheads.
>
>
> We hope the above discussions could lead to a better understanding of our theoretical contributions, and the effectiveness of the proposed methodologies.
>
> Best,
>
> Authors of paper 887

---

> > ### Author Response · Authors · 2021-11-23
> > **Sincerely expecting further discussions from Reviewer DHWs**
> >
> > Dear Reviewer DHWs:
> >
> > We want to thank you for the constructive comments in your review. As a follow-up on our responses, we would like to kindly remind that the discussion period is ending soon. We hope to use this open response period to discuss the paper to solve the concerns and improve the quality of our paper. Have you gotten a chance to read our responses above, which attempt to address all of your concerns?
> >
> > We have also improved the overall presentation of the paper, and included more visualizations you mentioned for 24-layer DeiT. Please kindly have a check on the updated PDF. We have made spectrum comparison between original and modified attention maps in Fig. 9 & 10 (Appendix G). We also directly plot the proportion of high-frequency components of feature signals in Fig. 11 (Appendix G). We hope these visualizations could persuasively demonstrate the effectiveness of our methods to you.
> >
> > We sincerely hope to have further discussions with you to see if our response solves the concerns. We would be more than happy to provide more information or clarification, should it be necessary. We hope our paper could receive a positive and fair assessment. We sincerely wish to bring our theoretical analysis and solution for deep ViT into the ViT community.
> >
> > Best,
> >
> > Authors of paper 887

---

> ### Author Response · Authors · 2021-11-26
> **Kindly expecting further discussions**
>
> Dear reviewer DHWs:
>
> We noted that the main concern you left in your review is "Can the authors provide more experiments on big models to further verify the effectiveness of the proposed method". In response to this, we have executed experiments correspondingly, and the new results are presented in section 5.1 table 1 of the updated PDF. Could you please kindly check them, and our responses above?
>
> We also would like to kindly note that the discussion phase is ending soon, we would like to see if there's updated evaluation of our work based on the new results.
>
> Looking forward to your reply!
>
> Best,
>
> Authors of paper 887

---

> > ### Comment · Reviewer_DHWs · 2021-11-27
> > **follow up**
> >
> > thanks for your response and additional experiments. Another question can you present FLOPs of your model and compare with the alternatives.

---

> > > ### Author Response · Authors · 2021-11-28
> > > **Reply to Reviewer DHWs**
> > >
> > > Dear Reviewer DHWs,
> > >
> > > Thanks for your reply. In regards to your comments, we provide the FLOPs in the following table:
> > >
> > > | Models | # Layer | # Params (M) | FLOPs (G) |
> > >  ------------- | ------------- | ------------- | ------------- |
> > > | DeiT-S  | 12  | 22.0 | 4.57 |
> > > | DeiT-S + AttnScale | 12 | 22.0 | 4.57 |
> > > | DeiT-S + FeatScale | 12 | 22.0 | 4.57 |
> > > | DeiT-S  | 24  | 43.3 | 9.09 |
> > > | DeiT-S + AttnScale | 24 | 43.3 | 9.10 |
> > > | DeiT-S + FeatScale | 24 | 43.4 | 9.10 |
> > > | CaiT-S  | 24  | 46.9 | 9.33 |
> > > | CaiT-S + AttnScale | 24 | 46.9 | 9.34 |
> > > | CaiT-S + FeatScale | 24 | 46.9 | 9.34 |
> > > | Swin-S  | 24  | 49.6 | 8.74 |
> > > | Swin-S + AttnScale | 24 | 49.6 | 8.75 |
> > > | Swin-S + FeatScale | 24 | 49.6 | 8.75 |
> > >
> > > We observe that our methods bring little FLOGs overheads, while contributing a significant performance gain. We will add these FLOPs metrics into our Table 1 in the camera-ready version if accepted.
> > >
> > > Best,
> > >
> > > Authors of paper 887

---

> > > ### Author Response · Authors · 2021-11-29
> > > **Thank you**
> > >
> > > Dear reviewer DHWs:
> > >
> > > Thank you for the precious endorsement and review!
> > >
> > > Best,
> > >
> > > Authors of paper 887

---

### Official Review · Reviewer_D5xX · 2021-11-05

**Correctness:** 3
**Technical Novelty And Significance:** 2
**Empirical Novelty And Significance:** 2
**Recommendation:** 6
**Confidence:** 2

**Main Review:**

Overall, this paper proposes a novel idea of thinking self-attention weight matrix as a low pass filter. This explains the difficulty of convergence for deeper tranformer models. The proposed methods alleviates the low-passing representation of the transformers. The experiments also prove the superior performance using proposed modules. However, the experiments are not sufficient, the improvement is relatively minor, and it doesn't show whether the proposed approach can help deeper transformers converging. It would be very interesting to see it is helpful for deep models.

**Summary Of The Paper:**

This paper first analysis the difficulty of going deeper using transformer models. The challenge is that the mechanism of self-attention is applying a low-pass filter.  Keep stacking self-attention layers will lose the feature expressive power and only preserve the DC bias.  To alleviate this issue, this paper propose two modules to make the MHA to be an all pass filter. First module AttenScale combines the high-pass component and FeatScale reweights the featuremap. The experimental results demonstrate the improvement on accuracy for DeiT and CaiT when adopting the proposed method.

**Summary Of The Review:**

Interesting and novel idea and good mathematical analysis. May need more experiments to demonstrate the claims in the paper.

---

> ### Author Response · Authors · 2021-11-14
> **Reply to Reviewer D5xX**
>
> Dear reviewer D5xX:
>
> We highly appreciate your acknowledgement of this work. We have provided detailed responses to your concerns point by point below, hoping them to help to build up further understanding.
>
> ***Q1.*** The experiments are not sufficient, the improvement is relatively minor.
>
> ***Reply:***
>
> We are confident that our improvement is not only practically significant but also has stronger theoretical potential. Specifically, comparing with the state-of-the-art ViT works, our FeatScale boosts 12-layer DeiT by 1.1%, while accuracy gain of re-attention in [1] boosts ~0.7%, LayerScale in [2] boosts ~0.6%, Late class token insertion in [2] boost ~0.7%. Besides showing stronger performance than the SOTA works, our techniques can also seamlessly cooperate with all of the existing architectures and add-ons, and even further boost their performances. This means that by simply plugging in our proposed techniques, every tested ViT gets boosted by >= 0.5% performance gain, with almost no additional complexity and memory overheads.
>
> Besides the originally reported performance improvements, we have updated two other experiments to make our model even stronger:
>
> 1. On Swin-S, our AttnScale and FeatScale achieve 0.4% and 0.5% performance gain, respectively. These improvements already surpass DiversePatch (~0.3%) [3].
>
> 2. We train a 24-layer CaiT-S with our approaches from scratch. Both AttnScale and FeatScale elevate accuracy by 0.6%. This makes CaiT-S (47M param)  even comparable to CaiT-M (186M param) with only hundreds of parameter increments.
>
> [1] Zhou et al., DeepViT: Towards Deeper Vision Transformer
>
> [2] Touvron et al., Going deeper with Image Transformers
>
> [3] Gong et al. Vision Transformers with Patch Diversification
>
> ***Q2.*** It doesn't show whether the proposed approach can help deeper transformers converging.
>
> ***Reply:***
> We appreciate your comments, and we agree that showing model convergence is an important aspect. However, what we claimed to address here is not the training convergence, but the over-smoothening rate w.r.t. the current layer depth (the deeper, the more over-smoothing).
>
> As clearly stated in our Theorem 3, the feature representation converges to its low-frequency components with an exponential rate, with regard to depth. Our theorem 3 described the downgradation of the expressiveness of ViT, instead of the convergence of model parameters.
>
> To show how our approaches can prevent this undesirable over-smoothing, we plot two commonly used metrics to illustrate the feature/attention[1][2] similarity in Figure 4. Both features/attention similarity is reduced by our methods, which supports our argument.
>
> [1] Zhou et al., DeepViT: Towards Deeper Vision Transformer
>
> [2] Gong et al. Vision Transformers with Patch Diversification
>
> Best,
>
> Authors of paper 887

---

> > ### Author Response · Authors · 2021-11-23
> > **Sincerely expecting further discussions from Reviewer D5xX**
> >
> > Dear Reviewer D5xX:
> >
> > We want to thank you here, again, for the constructive comments and acknowledgment of this paper. We have provided more experiment results on Swin and CaiT (train from scratch), and have demonstrated the good performance of the proposed methods (Table 1). We have also included more visualization in our appendix to further support the effectiveness of our method (Appendix G). Could you please kindly check the updated PDF and our response above, to see if your concerns are solved?
> >
> > We would be more than happy to provide more information or clarification, should it be necessary. We hope our paper could receive a positive and fair assessment, to help bring this mathematical anatomy and theoretical solutions to ViT community.
> >
> > Best,
> >
> > Authors of paper 887

---

### Author Response · Authors · 2021-11-23
**General Response**

Dear reviewers and AC:

We thank all the reviewers for their time and efforts on review. According to their precious feedback, we have revised our paper and uploaded the latest PDF. The change-log is summarized as below.

- A  couple of modifications have been included into the current version, as highlighted in the main text. Most of them concentrate on the experiment and appendix. The theoretical part and technical idea stay all the same.

- We thank all the reviewers for their suggestions on providing more convincing experiments. To this end, we have included more results on CaiT and Swin in Table 1, and deferred the fine-tuning results into the Appendix G.1.

- We thank DHWs for his/her suggestions on providing more visualization. We have replaced Fig. 4 in main text to show the attention/feature similarity for deep 24-layer model, and defer the original one (for 12-layer) to Appendix Fig. 12. We also provide direct visualization to the spectrum of modified attention map (Fig. 10) and the layer-wise dynamics of features’ high-frequency component (Fig. 11).

- We would appreciate it if all reviewers could please take a look and finalize their assessments on our work, hopefully more positively. We trust the reviewer and AC discussion would eventually lead to an informed and fair decision, and we thank everyone again for the valued efforts!

Best,

Paper 887 Authors

---

### Decision · Program_Chairs · 2022-01-20

**Decision:**

Accept (Poster)

**Comment:**

The paper analyses the frequency filtering properties of self-attention in vision architectures, shows that it mainly acts as a low-pass filter, and proposes fixes that allow to better preserve the higher frequencies. These fixes yield moderate classification accuracy gains (~0.5-1%) for several existing attention-based architectures.

The reviewers are quite borderline about the paper, but after considering the authors' responses lean towards acceptance. Pros include interesting and novel analysis and sound model improvements leading to non-trivial empirical gains. The main con is that the experimental results are fine, but not outstanding.

Overall, I recommend acceptance. Empirical results are indeed good but not outstanding, but the theoretical analysis is interesting and it is good to see that it leads to actionable insights on the model design side that actually help in practice - even is not by a huge amount. One part that in my opinion is confusing (and might have been confusing to the reviewers too) is that the title seems to suggest the paper will present very deep vision transformers while it does not. Adding deeper models or adjusting the title would help here.